# Isotope Geochemistry of the Shenshuitan Gold Deposit within the Wulonggou Gold Field in the Eastern Kunlun Orogen, Northwest China: Implications for Metallogeny

**Xuan Zhou [1], Tong Pan [2], Qing-Feng Ding [1,\*], Long Cheng [1], Kai Song [1], Fei Liu [1] and Yang Gao [1]**

[1] College of Earth Sciences, Jilin University, Changchun 130061, China; xuanzhou21@mails.jlu.edu.cn (X.Z.); wshdzhr@163.com (L.C.); songkai19@mails.jlu.edu.cn (K.S.); liufei9@smedi.com (F.L.); yanggao@jlu.edu.cn (Y.G.)

[2] Bureau of Geological Exploration and Development of Qinghai Province, Xining 810008, China; pant66@163.com

\* Correspondence: dingqf@jlu.edu.cn; Tel.: +86-4318-8502-278

**Abstract:** The Shenshuitan gold deposit is located within the Eastern Kunlun Orogen in northwest China. The gold mineralization here occurs primarily within the ductile fault XI. The sulfide mineral assemblage is dominated by pyrite and arsenopyrite, with minor pyrrhotite, chalcopyrite, galena, and sphalerite. Host rocks predominantly consist of Ordovician silicic slate and late Silurian granites, and their alterations include silicification and sericitization. The measured $\delta D$ and $\delta^{18}O$ values of quartz and sericite range from $-113.9‰$ to $-93.1‰$ and from $4.6‰$ to $12.0‰$, respectively. Bulk and in situ $\delta^{34}S$ values of sulfides range from $-7.3‰$ to $+9.6‰$ and from $-3.92‰$ to $11.04‰$, respectively. Lead isotope compositions of sulfides show $^{206}Pb/^{204}Pb$ ratios from 18.071 to 19.341, $^{207}Pb/^{204}Pb$ ratios from 15.530 to 15.67, and $^{208}Pb/^{204}Pb$ ratios from 37.908 to 38.702. Collectively, the isotope (H, O, S, and Pb) geochemistry suggests that the ore-forming fluids were of a metamorphic origin mixed with meteoric water and that the sulfur and lead were sourced from a mixture of host rocks and original ore-forming metamorphic fluids. Lastly, this deposit can be classified as an orogenic gold deposit associated with the final collision between the Bayan Har–Songpanganzi Terrane and the Eastern Kunlun Orogen during the Later Triassic.

**Keywords:** isotope geochemistry; ore-forming fluids; sulfur and lead sources; orogenic gold deposit; Shenshuitan gold deposit; Wulonggou gold field; Eastern Kunlun Orogen



## 1. Introduction

The Eastern Kunlun Orogen (EKO) has a complicated geodynamic evolutionary history and is an important ore belt where numerous ore prospecting projects have been implemented by the China Geological Survey since 1999. Thus far, there have been more than 140 mineral occurrences associated with metals such as Au, Fe, Cu, Pb, Zn, W, Mo, Ni, Co, and Sb in the EKO and the Bayan Har–Songpanganzi Terrane (BH–SG) on the south of the EKO, and more than 26 of them have been developed into mineral deposits of different sizes [1,2]. In particular, the EKO hosts numerous gold deposits, best represented by those in the Wulonggou gold field and Gouli gold field, as well as the Kaihuangbei gold deposit [3–9].

The Wulonggou gold field within the EKO is characterized by extensive granitic magmatism, ductile faults, and hydrothermal gold mineralizations [4,10,11]. Hydrothermal gold mineralizations are predominantly controlled by the three northwest-trending ductile fault zones, i.e., Yanjingou (fault I to fault XI), Yingshigou–Hongqigou (fault VII to fault XI), and Sandaoliang–Kushuiquan (fault XII to fault XV) [4,9,10,12–14]. The earliest-found Yanjingou (also known as Shihuigou) gold deposit is controlled by fault III within the Yanjingou fault zone, while the Shenshuitan, Danshuigou, and Hongqigou gold deposits are controlled

by faults VII, IX, and X–XI within the Yingshigou–Hongqigou fault zone, respectively. The aforementioned four deposits are the most important gold deposits in the Wulonggou gold field [15]. The Yanjingou gold deposit was discovered earlier than the others; thus, the majority of previous studies have focused on this deposit, including its geological characteristics [16–20], ore-forming fluids [16,18,21,22], ore-controlling structures [12,13,16,23], ore genesis [9,14,16,18–20,22,24,25], and ore-forming ages [4,9,18,19,24,26,27]. The relative consensus amongst scientists is that the Yanjingou gold deposit belongs to an orogenic gold deposit, and ore types within it primarily consist of quartz vein and phyllic rock [9,17–19,21,22,25,28]. By 2010, the gold reserve increased to 43.7 tons for gold deposits within the Wulonggou gold field, mostly because of the discovery of the Shenshuitan gold deposit [15]. Thus far, few geologists have carried out geological research on the Shenshuitan gold deposit. We previously reported that the ore-forming age of the Shenshuitan gold deposit is later than 215 Ma [4]. Zhang et al. (2017) argued the existence of a possible genetic relationship between Shenshuitan gold mineralization and post-collisional magmatism in the EKO [9]. Zhang (2018) believes that it belongs to an orogenic gold deposit considering research pertaining to fluid inclusion and H, O, S, and Pb isotopes [22]. Recently, Cheng (2020) reported H, O, S, and Pb isotope compositions of gold-bearing quartz sulfide veins in the Hongqigou gold deposit [29], which is located 1 km northeast of the Shenshuitan gold deposit and is also controlled by the Yingshigou–Hongqigou fault zone, and it was argued that the Hongqigou gold deposit also belongs to an orogenic gold deposit. Orogenic gold deposits usually show similar geotectonic settings of orogeny but variable origins of ore-forming fluids and sources of metals [30,31]. At present, the origins of ore-forming fluids and sources of metals responsible for gold mineralizations in the Shenshuitan gold deposit remain controversial.

In this paper, we report a systematic study of the H, O, S, and Pb isotope compositions of gold-bearing ores in the Shenshuitan gold deposit. Comparing our data with published data on the Shenshuitan gold deposit, as well as published data on the nearby Hongqigou gold deposit, we can characterize the origins of ore-forming fluids, sources of metals, and the ore genesis of the Shenshuitan gold deposit. Moreover, our study contributes to a better understanding of this type of gold mineralization in the whole Wulonggou gold field, which might be helpful for further gold exploration in the EKO.

## 2. Geological Characteristics

### 2.1. Geological Background

The EKO, which is bound by the Qaidam block to the north and the BH–SG to the south, is commonly divided into the fault-bound North Kunlun belt (NKL), Middle Kunlun belt (MKL), and South Kunlun belt (SKL) (Figure 1b) [32]. The ophiolite belt, along the Middle Kunlun Fault (M.KLF) between MKL and SKL and that along the South Kunlun–Aryan Maqin Fault (S.KLF–AMF) between the SKL and BH–SG, represents a suture zone of the Neoproterozoic to Early Devonian Proto-Tethys and the Carboniferous to Late Triassic Paleo-Tethys oceans, respectively [32–34]. The EKO has a complex geological history with multiple stages of orogeny that are closely related to the consumption of the Proto-Tethys and Paleo-Tethys oceans [32–37]. Therefore, widespread stages of different granitic intrusions were emplaced within the EKO, related to the evolutions of the Proto-Tethys and Paleo-Tethys oceans (e.g., [4,9–11,38–43]).

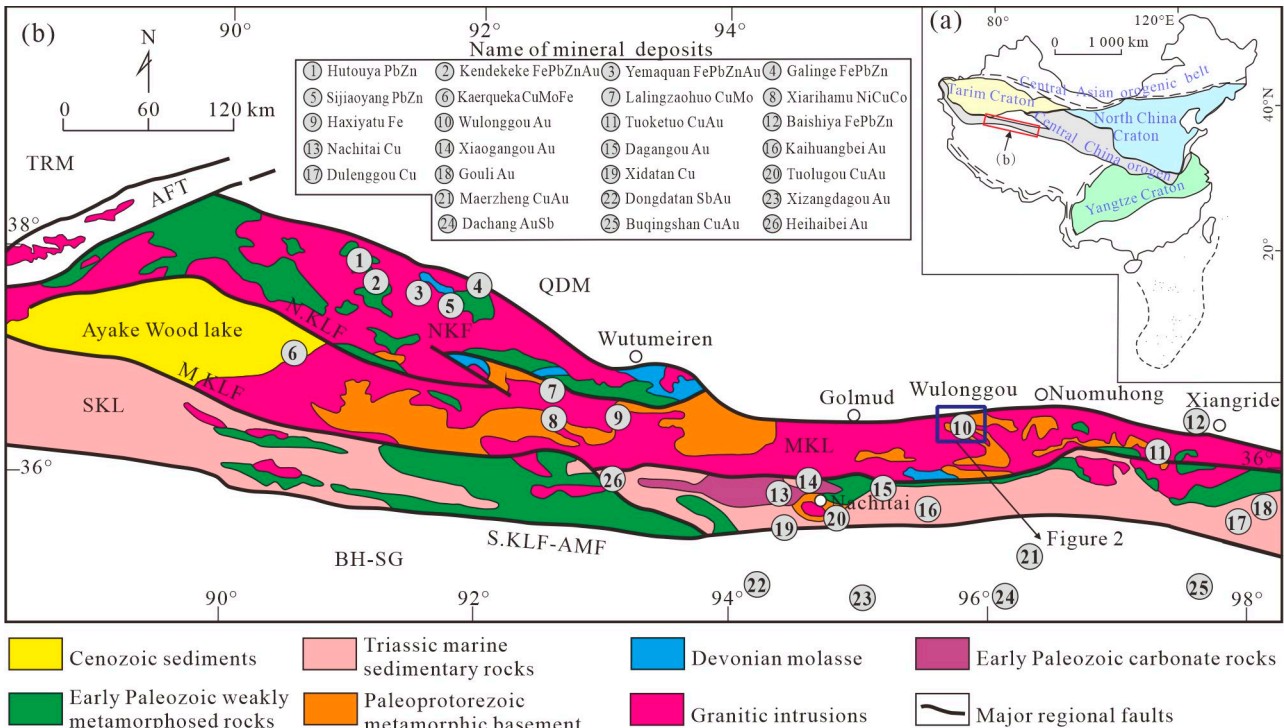

**Figure 1.** (**a**) Sketch map of China showing the tectonic location of the EKO, revised after [44]; (**b**) schematic geologic map of the EKO, simplified and modified after [45]. EKO: Eastern Kunlun Orogen; QDM: Qaidam Block; TRM: Tarim Basin; NKL: North Kunlun Belt; MKL: Middle Kunlun Belt; SKL: South Kunlun Belt; BH–SG: Bayan Har–Songpanganzi Terrane; ATF: Altyn Tagh Fault; N.KLF: North Kunlun Fault (South Qiman Tagh Fault); M.KLF: Middle Kunlun Fault; S.KLF–AMF: South Kunlun–Aryan Maqin Fault. Major mineral deposits in the EKO are after [1,2].

The Wulonggou gold field is tectonically located in the central segment of the MKL in the EKO (Figure 1b). The MKL is characterized by Early Cambrian–Early Devonian and widespread Late Permian–Early Triassic granitic rocks (e.g., [38,46,47]). These granitic rocks were emplaced in a Proterozoic metamorphic complex basement, which dominantly consists of Precambrian gneiss to schist to phyllite [33,48].

*2.2. Ore Deposit Geology*

Throughout the entire Wulonggou gold field, old metasedimentary rocks consist of the Paleoproterozoic Jinshuikou Group, Mesoproterozoic Xiaomiao Formation, and Neoproterozoic Qiujidonggou Formation (Figure 2) [15].

The area is also characterized by widespread granitic rocks, which include Cambrian, Silurian, Devonian, Permian, and Triassic granites [4,10,15]. Previous studies have shown that the Early–Middle Triassic (244–248 Ma [4,10]) granites and the Late Silurian to Early Devonian (416–420 Ma [11,45,49,50]) granites are the most developed intrusive host rocks of the gold mineralization. Three NW-trending ductile fault zones (Figure 2), i.e., Yanjingou (fault I–fault VI), Yingshigou–Hongqigou (fault VII–fault XI), and Sandaoliang–Kushuiquan (fault XII–fault XV), host the Shenshuitan, Yanjingou, Hongqigou, and Danshuigou gold deposits, as well as numerous gold occurrences throughout the entire Wulonggou gold field [9,12,13].

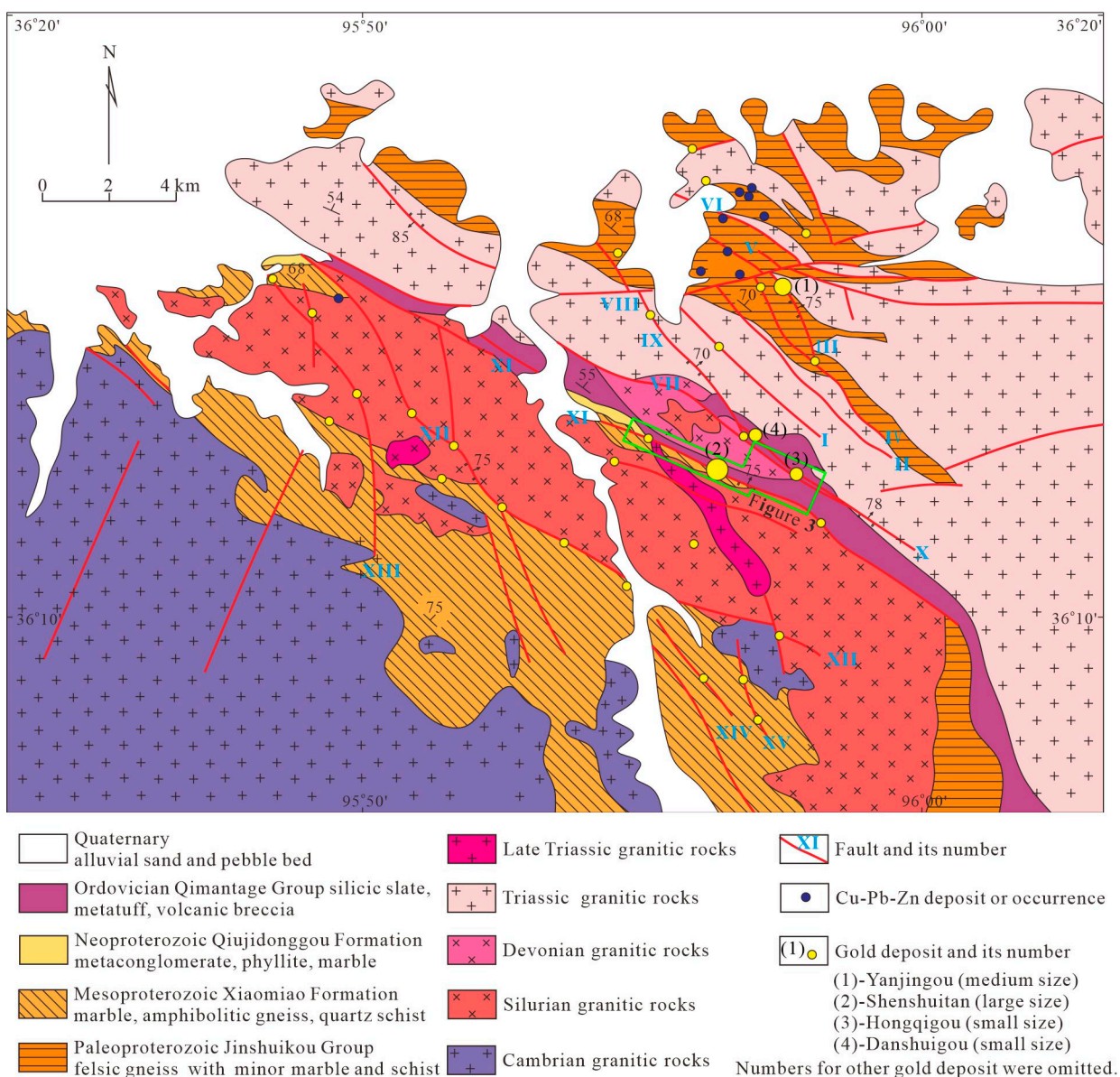

**Figure 2.** Geologic map of the Wulonggou area in the EKO (modified after [10]).

Fault XI, about 4 km long and 30–100 m wide, strikes southeast and generally dips steeply to the northeast in the central part of the Wulonggou gold field [15]. The large-scale Shenshuitan gold deposit, including three ore districts, i.e., Shuizhadonggou, Shenshuitan, and Huanglonggou, is controlled by the east segment of fault XI within the Yingshigou–Hongqigou fault zone (Figure 3a). Numerous areas of Ordovician Qimantage Group silicic slate, metatuff, and volcanic breccia strike across the central part of the Shenshuitan mining area from northwest to southeast. Mesoproterozoic Xiaomiao Formation marble, amphibolite gneiss, and quartz schist are mostly located in the west segment of the mining area, while minor Neoproterozoic Qiujidougou Formation meta-conglomerate and phyllite occur in the east segment (Figure 3a) [9,15]. Extensive Late Silurian granites were emplaced in the south of the mining area, and the Early Devonian granites were intruded in the north. A few small-scale pre-Devonian granites in the south of the mining area were intruded or trapped by the Late Silurian granites. Several pre-ore dioritic dykes (e.g., Huanglonggou diorite, 215 Ma after [4]) and a pre-ore granite stock (i.e., Huanglonggou granite, 219 Ma after [51] or 221 Ma after [9]) were intruded in the central part of the mining area (Figure 3a). Fault XI cuts all sedimentary rocks and all intrusions, including the latest Huanglonggou granite and Huanglonggou diorite in the Shenshuitan mining area; therefore, we argue that

the gold mineralizations in the Shenshuitan gold deposit occurred after these Late Triassic (e.g., 215 Ma) magmatic intrusions [4].

**Figure 3.** Geologic map of the Shenshuitan and Hongqigou gold deposits in the Wulonggou gold field (simplified and revised after [15]). (**a**) The Shenshuitan, Shuizhadonggou, and Huanglonggou gold deposits. (**b**) The Hongqigou and Heishigou gold deposits.

In the Shenshuitan gold deposit, fault XI developed along the contact between Qimantage Group silicic slate and Late Silurian granite. Such contact consists of a relatively sharp upper fault plane and a gradational lower fault plane. The faulted zone of fault XI is commonly filled with granitic breccias (Figure 4). Furthermore, extensive hydrothermal alterations are developed along and within the faulted zone of fault XI (Figure 4). Alterations are mostly composed of sericitization, silicification, and pyritization, with minor chloritization and carbonatization [9,15], belonging to a typical phyllic zone. Usually, sericitization, silicification, and pyritization exceed the lower fault plane and spread into the Late Silurian granite footwall, while only some pyritization can exceed the upper fault plane and develop in the Qimantage group silicic slate of the hanging wall (Figure 4). There are more than 120 gold ore bodies within the phyllic zone in the Shenshuitan gold deposit, and the ore bodies are mostly veined, lenticular, or striped, with the same occurrence as fault XI, dipping 355–40° with a dip angle of 60–85° [15]). Individual ore bodies are mostly 100–200 m long, 1–15 m wide, and 0–300 m deep. They are generally discontinuous and characterized by branches and bulges along the strike and dip. Only LM8, LM11, LM18, and LM23 ore bodies are very important, with a length of more than 200 m and an average gold grade of 6.74 ppm [15].

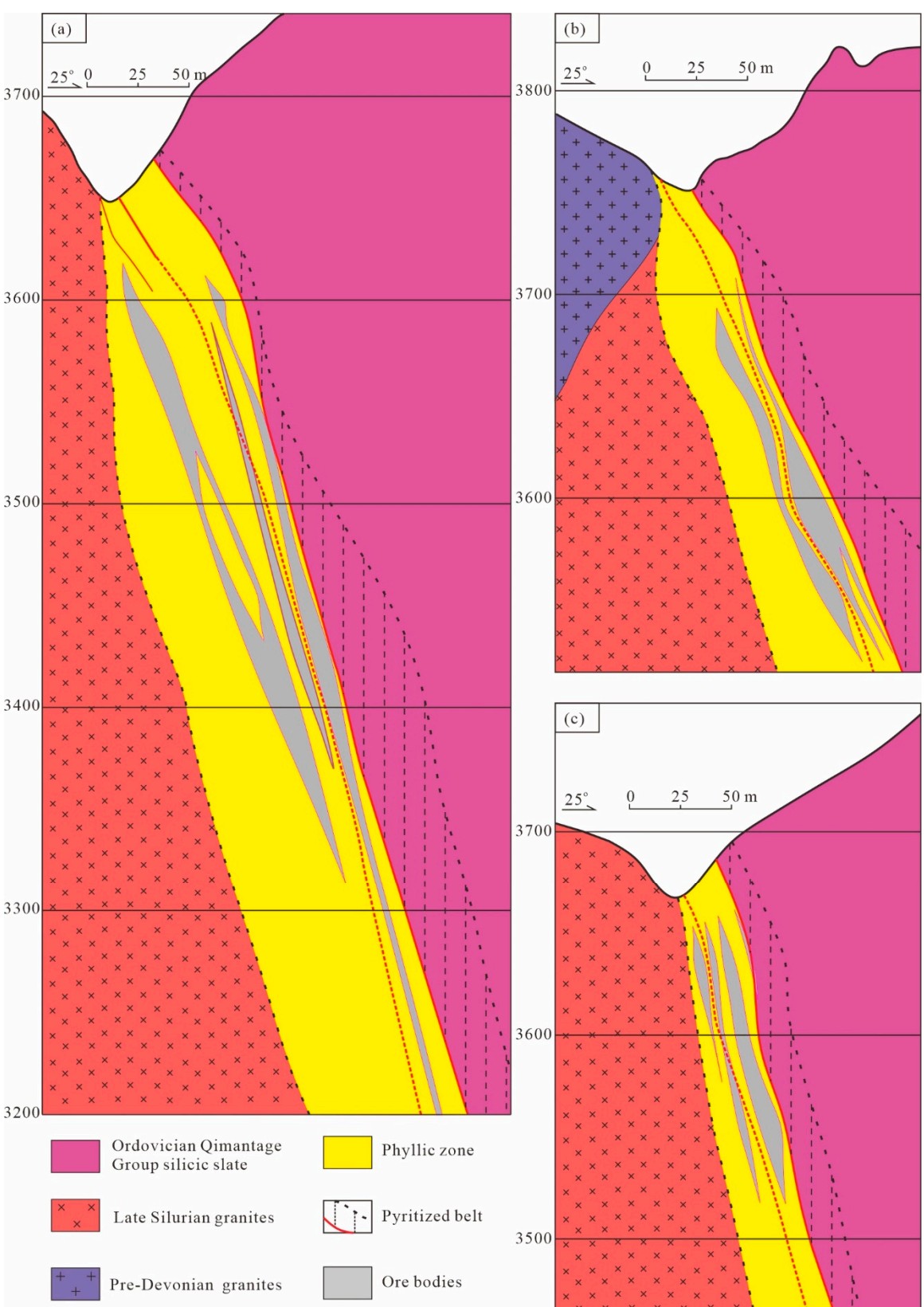

**Figure 4.** Typical sections of the Shenshuitan gold deposit (simplified and revised from [15]). (**a**) Prospecting line 11, (**b**) prospecting line 21, and (**c**) prospecting line 29.

The dominant ore types within ore bodies are phyllic rocks in the Shenshuitan gold deposit [15]. Sulfide disseminations in the phyllic zone consist of pyrite and arsenopyrite,

with minor native gold, pyrrhotite, chalcopyrite, galena, and sphalerite, with contents in auriferous ores of <5% (Figure 5). Micro-native gold and electrum are mostly trapped in pyrite and arsenopyrite [15]. The paragenetic sequences can be divided into two major stages, i.e., the hydrothermal stage (including quartz–sulfide epoch and calcite–sulfide epoch) and the supergene stage (Figure 6). The quartz, sericite, pyrite, and arsenopyrite assembly represents the quartz–sulfide epoch of ores within faulted and brecciated granite along the faulted zone (ore subtype 1, Figure 5a–d) or ores in the pyritized silicic slate within the hanging wall (ore subtype 2, Figure 5g–i), and the quartz, calcite, pyrite, galena, sphalerite, and chalcopyrite assembly (ore subtype 3, Figure 5e,f) represents the calcite–sulfide epoch of ores that replaces the quartz–sulfide epoch of ores. The supergene stage predominantly consists of hematite, jarosite, and quartz, which resulted from the replacement of the hydrothermal stage of primary phyllic ores (Figure 6).

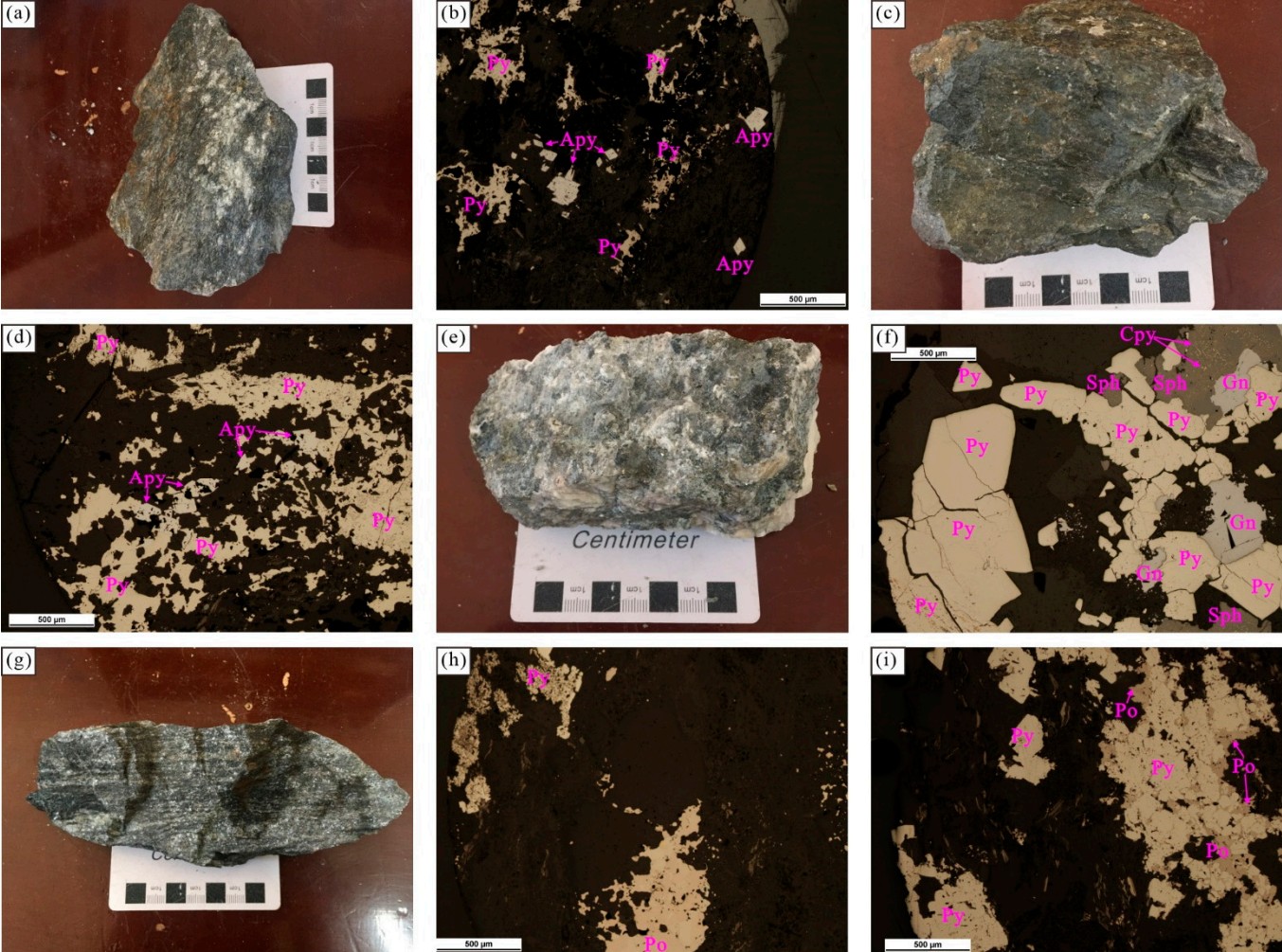

**Figure 5.** Photographs and photomicrographs showing the morphology and textural features of the ores in the Shenshuitan gold deposit. (**a**) Photograph and (**b**) photomicrograph of sample XI-3390NCM29-B3 of ore subtype 1; (**c**) photograph and (**d**) photomicrograph of sample XI-3390NCM29-B4 of ore subtype 1; (**e**) photograph and (**f**) photomicrograph of sample XI-3390NCM29-B5 of ore subtype 3 within ore subtype 1; (**g**) photograph and (**h,i**) photomicrograph of sample XI-3390NCM29-B6 in ore subtype 2. Apy, Arsenopyrite; Py, pyrite; Po, pyrrhotite; Gn, galena; Sph, sphalerite; Cpy, chalcopyrite.

| Minerals \ Stages | Hydrothermal stage — Quartz–sulfide epoch | Hydrothermal stage — Calcite–sulfide epoch | Supergene stage |
|---|---|---|---|
| Native gold | ▬ | | |
| Pyrite | ▬▬ (thick) | ▬ | |
| Arsenopyrite | ▬ | | |
| Pyrrhotite | ▬ | | |
| Sphalerite | ▬ | ▬ (thick) | |
| Chalcopyrite | ▬ (faint) | ▬ | |
| Galena | | ▬ | |
| Jarosite | | | ▬ |
| Haematite | | | ▬ |
| Quartz | ▬ | ▬ | |
| Sericite | ▬ | | |
| Calcite | | ▬ (faint) | |

**Figure 6.** Paragenetic sequences of minerals in different ore-forming stages in the Shenshuitan gold deposit.

The Hongqigou gold deposit is located 1 km northeast of the Shenshuitan gold deposit. The gold bodies in the Hongqigou gold deposit are controlled by the NW-striking faults X, VII, and IX within the Yingshigou–Hongqigou ductile fault zone (Figure 3b). The sedimentary rock is composed of Paleoproterozoic Jinshuikou Group felsic gneiss in the north and Ordovician Qimantage Group silicic slate, metatuff, and volcanic breccia in the south (Figure 3b). Early Devonian granites and Pre-Devonian granites intruded the sedimentary rocks (Figure 3b). There are about 53 gold ore bodies within phyllic zones or quartz veins in the Hongqigou gold deposit. Ore bodies are mostly lenticular, veined, or striped, with the same occurrence as faults, dipping 304–68° with a dip angle of 4–85° [15]. They are generally discontinuous and characterized by branches and bulges along the strike and dip. Individual ore bodies are mostly small-scale. Only QM4, QM5, and QM8 are important and have a length of more than 200 m [15]. Ores are composed of dominant phyllic rock with some quartz veins [15,29,52].

## 3. Sampling and Analytical Methods

### 3.1. Sampling

Eighteen ore samples were collected for isotope analyses from tunnels on certain prospecting lines, i.e., prospecting lines 29, 27, 11, and 7, at a level of 3390 m underground in the Shenshuitan gold deposit (Figure 7). The ore subtype, wall rock, and mineral assembly of the ore samples are listed in Supplementary Materials Table S1. These 18 sulfide-bearing ore samples were all hand-picked, and corresponding sections were polished for optical observation. One sericite and six quartz mineral separates were isolated and collected from seven of the ore samples for oxygen and hydrogen isotope analyses (Supplementary Materials Table S2). Forty-one sulfide mineral separates were isolated and collected from all 18 ore samples for bulk sulfur isotope analysis (Supplementary Materials Table S3). Sections of nine relatively sulfide-enriched ore samples were polished for in situ sulfur isotope analysis (Supplementary Materials Table S4). Forty-one sulfide separates were simultaneously collected from the 18 ore samples for bulk lead isotope analysis (Supplementary Materials Table S5).

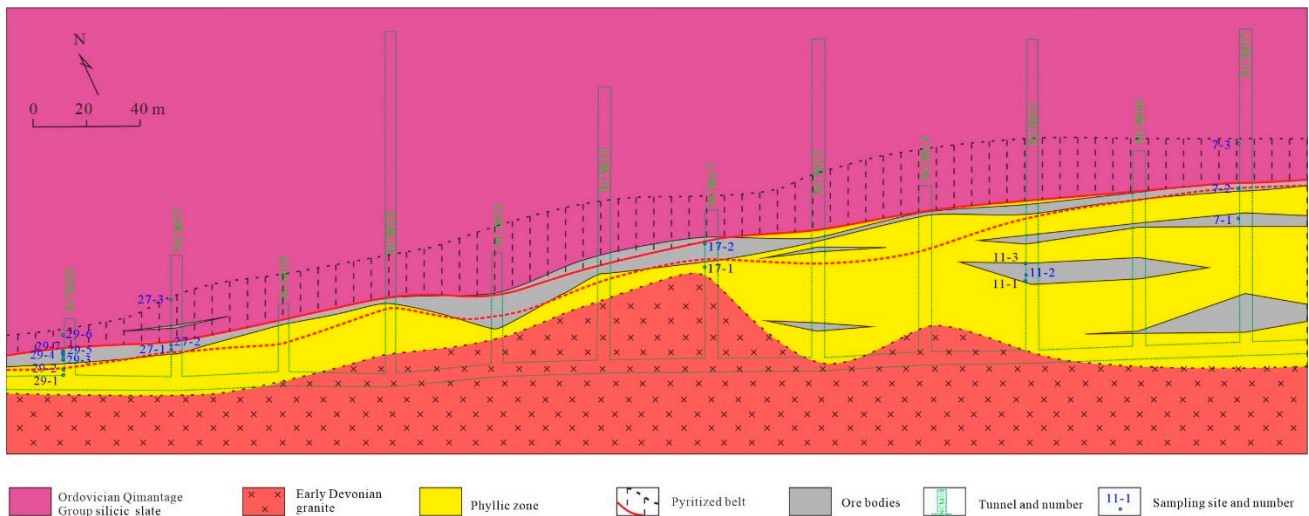

| | Ordovician Qimantage Group silicic slate | | Early Devonian granite | | Phyllic zone | | Pyritized belt | | Ore bodies | | Tunnel and number | 11-1 | Sampling site and number |

**Figure 7.** Geologic sketch map of Level 3390 m in the underground Shenshuitan gold deposit (simplified and modified after [52]); sampling sites and numbers are also shown ("XI-3390NCM" at the beginning of each sample name has been omitted, e.g., sampling site "XI-3390NCM29" has been abbreviated to "NCM29").

### 3.2. Oxygen and Hydrogen Isotope Analysis

Oxygen and hydrogen isotope analyses were carried out at the Analytical Laboratory of BRIUG (Beijing Research Institute of Uranium Geology, China National Nuclear Corporation (CNNC)), using methods described in [5]. Oxygen isotope analyses were carried out on 5 to 10 mg of quartz using a Thermo Fisher MAT253–EM via the bromine pentafluoride method described in [53]. The hydrogen isotope composition of quartz samples was determined by decrepitation of the fluid inclusions within them. The results are reported relative to VSMOW, and the analytical precisions were ±0.2‰ for $\delta^{18}O$ and ±1‰ for $\delta D$. Isotopic fractionation of oxygen between quartz and water was calculated using the equation $1000\ln\alpha = 3.38 \times 10^6/T^2 - 3.4$ [54]. Isotopic fractionation of oxygen between sericite and water was calculated using the equation $1000\ln\alpha = 3.39 \times 10^6/T^2 - 3.76$ [55,56], and isotopic fractionation of oxygen was calculated using the equation $1000\ln\alpha = -20$ [56,57].

### 3.3. Bulk Sulfur Isotope Analysis

Bulk sulfur isotope analyses of sulfide separates were carried out on 10 to 100 mg of sulfide minerals including pyrite, arsenopyrite, and pyrrhotite from 18 samples. The sulfur isotope compositions were determined by a Thermo Scientific Delta V Plus mass spectrometer at the Analytical Laboratory of BRIUG. Sulfur isotope compositions of sulfide minerals were measured using the conventional combustion method [58], as described in [59]. The results are reported as $\delta^{34}S$ relative to VCDT, and the analytical precision was ±0.2‰.

### 3.4. In Situ Multiple Sulfur Isotope Analysis

In situ sulfur isotope analysis for sulfide samples by SIMS (secondary ion mass spectrometry) was done at the CMCA (Center for Microscopy and Microanalysis), UWA (University of Western Australia) using the CAMECA IMS 1280 large-geometry ion microprobe following the procedures of [60]. Samples containing sulfide mineral assemblages were selected for analysis based on their assemblages determined through optical microscopy, using TESCAN VEGA3 SEM–BSE (scanning electron microscopy–back scatter electron) and X-Max 50 silicon drift EDS (energy-dispersive X-ray spectroscopy) analysis at CMCA, UWA. The SIMS analyses were completed on pyrite and pyrrhotite for the deposit, and

standards used for SIMS analysis were based on their sulfide phases, i.e., Sierra for pyrite and Alexo for pyrrhotite, which are defined in [60].

Sample preparation started with each polished block sample drilled using a drill press producing cylindrical shaped samples with a 3.3 mm diameter, before being mounted into a resin mount. Resin mounts containing six samples each were arranged based on their sulfide phases. Samples XI-3390NCM7-B1, XI-33907-B3, XI-339011-B2, XI-339027-B1, XI-339029-B3, XI-339029-B4, XI-339029-B5, XI-339029-B6, and XI-339029-B7 were mounted into four resin mounts (Supplementary Materials Table S4). The limited amount of sulfide standard incorporated into each standard block determined the sample arrangement depending on the standard block to maximize efficiency.

Sample mounts were trimmed using a precision saw to a thickness of 5 mm and then coated with a 30 nm gold coating for the SIMS analysis. Four sulfur isotopes, i.e., $^{32}S$, $^{33}S$, $^{34}S$, and $^{36}S$, were analyzed relative to VCDT. The ability of SIMS to analyze a spot $15 \times 15$ μm in size was an important element in this research, as many of the gold-associated sulfides were either very small or zoned. The detailed analytical procedures for the in situ sulfur isotope analysis by SIMS are described in [60,61]. The mass-independent fractionations of sulfur are denoted by $\Delta^{33}S$ (= $\delta^{33}S - 1000((1 + \delta^{34}S/1000)^{0.515} - 1)$) and $\Delta^{36}S$ (= $\delta^{36}S - 1000((1 + \delta^{34}S/1000)^{1.91} - 1)$) notation to represent the deviation between the isotopic ratios measured and those predicted according to mass-dependent fractionation [60,62].

### 3.5. Lead Isotope Analysis

Lead isotope analyses were conducted on an Isotopx Phoenix TIMS (thermal ionization mass spectrometer) at the Analytical Laboratory of BRIUG, using methods described in [63]. About $100-200$ mg of the sulfide separates were completely dissolved in an ultrapure acid mixture of $HF + HNO_3 + HClO_4$. After drying, the residue was redissolved in HCl and dried again. Next, the sample residue was dissolved in 0.5 mol/L HBr and then loaded into a column with 250 μL of AG 1-X8 anion resin to separate the Pb fraction. Finally, the residue was redissolved in HCl, yielding $PbCl_2$ of high purity. The $PbCl_2$ was placed on a Re filament together with a solution of silica gel and phosphoric acid before measurement using an Isotopx Phoenix TIMS. All measured isotopic ratios were corrected by the NBS 981 standard. Analytical errors for $^{206}Pb/^{204}Pb$, $^{207}Pb/^{204}Pb$, and $^{208}Pb/^{204}Pb$ ratios were <0.05%.

## 4. Results

### 4.1. Oxygen and Hydrogen Isotope Results

Quartz samples from phyllic rock ores of ore subtypes 1 and 2 showed a $\delta^{18}O_{quartz}$ range from 4.6‰ to 12.0‰, with an average of 10.0‰ (Supplementary Materials Table S2). It is difficult to calculate the precise $\delta^{18}O_{H2O}$ values of fluids that were in equilibrium with quartz due to a large variation in temperatures obtained from fluid inclusions (e.g., [64]). The homogenization temperatures of quartz in ores of ore subtypes 1 and 2 ranged from 141 °C to 355 °C, with an average of 262 °C in the Shenshuitan gold deposit [22]. Such a wide range of temperatures is inconsistent with the lack of significant metal zoning in the deposit, and the low-temperature homogenization measurements are from post-ore inclusions and/or inclusions that have undergone post-entrapment modification (e.g., [31]). Typically, the trapping temperature can be defined by the average homogenization temperature to calculate isotopic fractionation values. Using 262 °C as the trapping temperature, the $\delta^{18}O_{H2O}$ values of the fluids (−3.8‰ to 3.6‰) for fluid inclusions from the quartz samples were calculated and are shown in Supplementary Materials Table S2. Given that the trapping temperature was defined by the highest homogenization temperature of 355 °C, the calculated $\delta^{18}O_{H2O}$ values of the fluids have only a 3.2‰ increment and vary from −0.6 to 6.8‰ (Figure 8). Hydrogen isotope results for fluid inclusions from quartz samples are also shown in Supplementary Materials Table S2 with δD ranging from −113.9‰ to −103.6‰, with an average of −110.6‰ (Figure 8). The hydrogen–oxygen isotopes in

ore subtype 1 and ore subtype 2 were similar, although there is only one dataset for ore subtype 2.

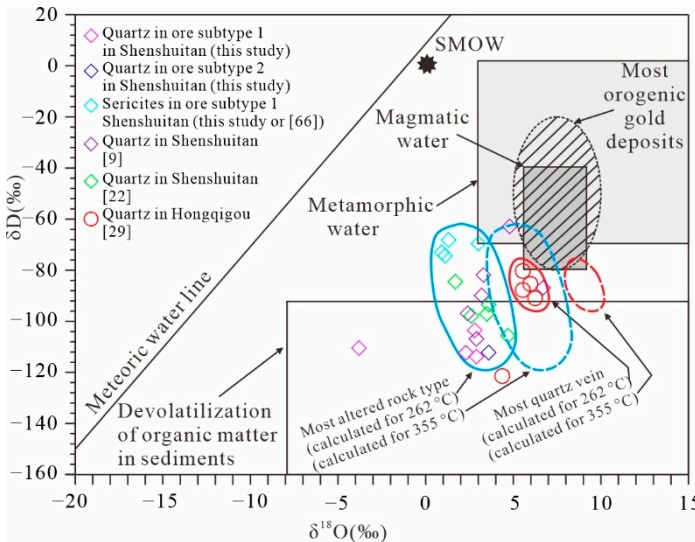

**Figure 8.** Plot of δD vs. $\delta^{18}O_{H2O}$ for ore-forming fluids from the Shenshuitan gold deposit. Magmatic, metamorphic, and organic (e.g., devolatilization of organic matter in sediments) water fields after [65]; the field for most orogenic gold deposits was revised after [64]. Previously published data are from [9,22,29,66]. SMOW, standard mean ocean water.

One sericite sample from auriferous ores showed a $\delta^{18}O$ value of 5.5‰ and a δD value of −93.1‰ (Supplementary Materials Table S2), both of which are within the range of three sericites reported by [66]. Using 262 °C as the trapping temperature, a $\delta^{18}O_{H2O}$ value of 0.9‰ and a δD value of −73.1‰ for the fluids in the sericite sample were calculated, as shown in Supplementary Materials Table S2.

*4.2. Bulk Sulfur Isotope Results*

The $\delta^{34}S_{VCDT}$ values of 41 sulfide separates from 18 ore samples from the Shenshuitan gold deposit ranged from −7.3‰ to 9.6‰, with a high variation and an average of 2.5‰ (Supplementary Materials Table S3). Different sulfides yielded different $\delta^{34}S_{VCDT}$ values (Figure 9a). It is known that a higher valence state of sulfur in a sulfide results in a larger $\delta^{34}S$ value of the sulfide in a closed paragenetic system. However, the $\delta^{34}S$ values of all sulfides exhibited large variations, and the range of $\delta^{34}S$ values of different sulfides was not comparable, which would typically indicate an open system. Pyrite has a highly variable range of $\delta^{34}S_{VCDT}$ values, while other sulfides yield a relatively narrow range, perhaps because pyrite is the most common sulfide, even in barren host rocks.

The range of $\delta^{34}S_{VCDT}$ values of sulfides from different ore subtypes varied, with ore subtype 1 exhibiting a range from −7.3‰ to 6.3‰ (Figure 9b), ore subtype 2 exhibiting a range from 2.8‰ to 9.6‰ (Figure 9c), and ore subtype 3 exhibiting a range from 3.0‰ to 5.7‰ (Figure 9d). The extent of $\delta^{34}S_{VCDT}$ values of ore subtype 1 spanned the range of ore subtype 3 values, and the range of ore subtype 2 values was slightly higher compared to the values of the other two ore subtypes (Figure 9f). Ore subtype 1 had highly variable $\delta^{34}S_{VCDT}$ values, likely because it was the most complicated mixture of ore-forming fluid and host rocks. Furthermore, our data spanned the previously published $\delta^{34}S_{VCDT}$ value ranges, i.e., 1.3‰ to 6.9‰ [9] and 0.4‰ to 6.9‰ [67]. However, the majority of $\delta^{34}S_{VCDT}$ values for phyllic rock ores in the Shenshuitan gold deposit (this study, [9,67]) are apparently lower than the majority of $\delta^{34}S_{VCDT}$ values for quartz veins in the Hongqigou gold deposit [29] (Figure 9e,f).

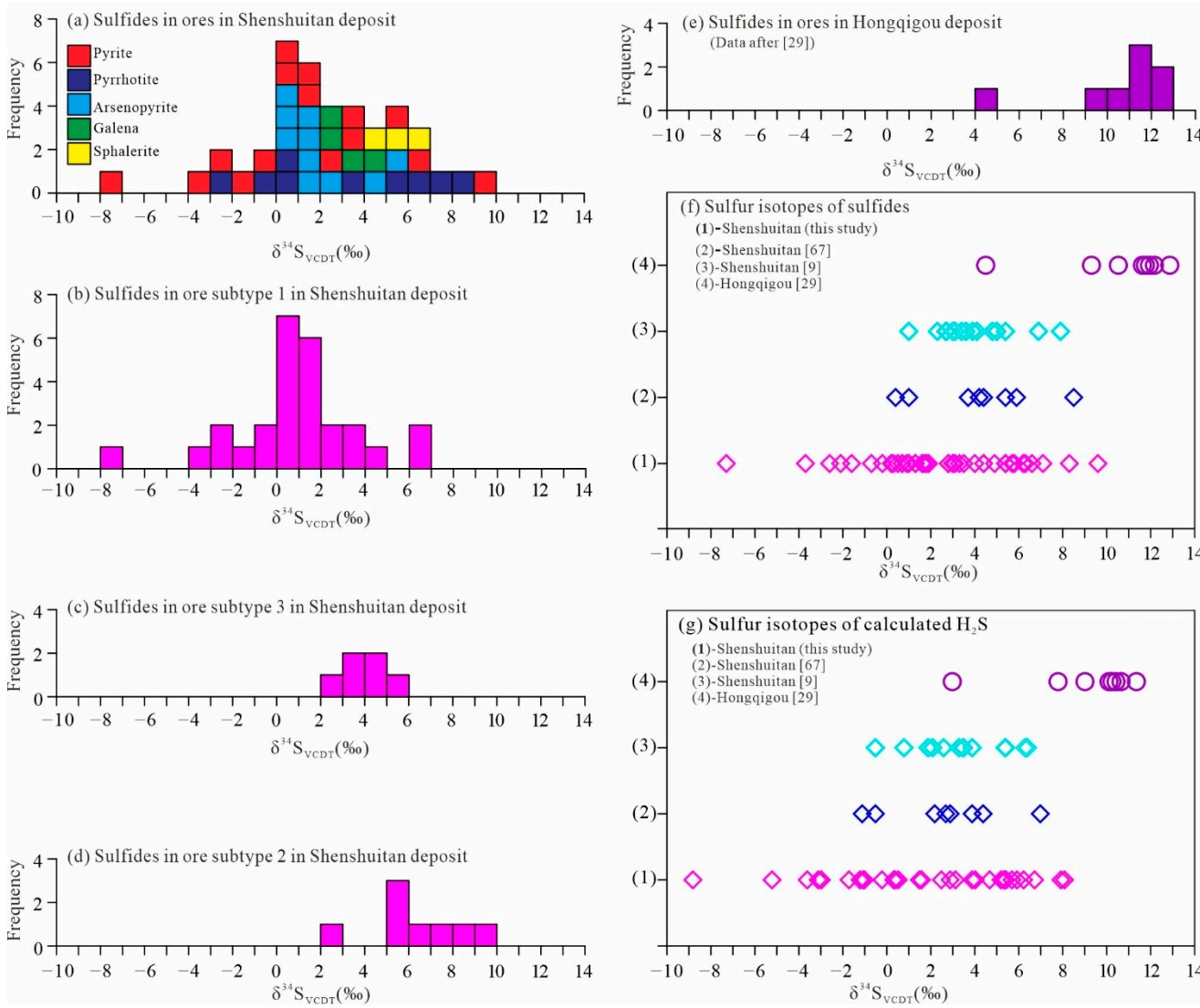

**Figure 9.** Bulk sulfide isotopes of sulfides in ores from the Shenshuitan deposit. (**a**) Histogram of sulfur isotopes for each sulfide; (**b**) histogram of sulfur isotopes for sulfides in ore subtype 1; (**c**) histogram of sulfur isotopes for sulfides in ore subtype 2; (**d**) histogram of sulfur isotopes for sulfides in ore subtype 3; (**e**) histogram of sulfur isotopes for sulfides from ores in the Hongqigou deposit; (**f**) plot of sulfur isotopes for sulfides in the Shenshuitan deposit compared with those in the Hongqigou deposit; (**g**) plot of calculated $H_2S$ isotopes in the Shenshuitan deposit compared with those in the Hongqigou deposit. Previously published data are from [9,29,67].

### 4.3. In Situ Multiple Sulfur Isotope Results

For comparison with the bulk sulfur isotope compositions, in situ multiple sulfur isotope analysis targeted two sulfide phases (pyrite and pyrrhotite) in nine phyllic rock ore samples from the Shenshuitan gold deposit. A summary of the data is reported in Supplementary Materials Table S4. All phases yielded $\delta^{34}S_{CDT}$ values ranging from $-3.92‰$ to $11.04‰$, with a mean of $3.45‰ \pm 0.22‰$ ($n = 273$; 2SD on the mean), $\delta^{33}S_{CDT}$ values ranging from $-1.99‰$ to $5.68‰$ with a mean of $1.75‰ \pm 0.17‰$ ($n = 273$; 2SD on the mean), and $\delta^{36}S_{CDT}$ values ranging from $-8.16‰$ to $21.12‰$ with a mean of $6.26‰ \pm 0.97‰$ ($n = 273$; 2SD on the mean). In all samples, pyrites yielded $\delta^{34}S$ values ranging from $-3.95‰$ to $9.94‰$ ($n = 224$), while pyrrhotite yielded $\delta^{34}S$ values ranging from $0.52‰$ to $11.04‰$ ($n = 49$) (Figure 10a). Lastly, in situ sulfur isotopes from different spots on individual samples were relatively stable; furthermore, in situ sulfur isotopes for each sample were very

similar to the bulk sulfur isotope composition of each sample (Figure 10b). It is obvious that in situ sulfur analyses are comparable and similar to the bulk sulfur isotope analyses.

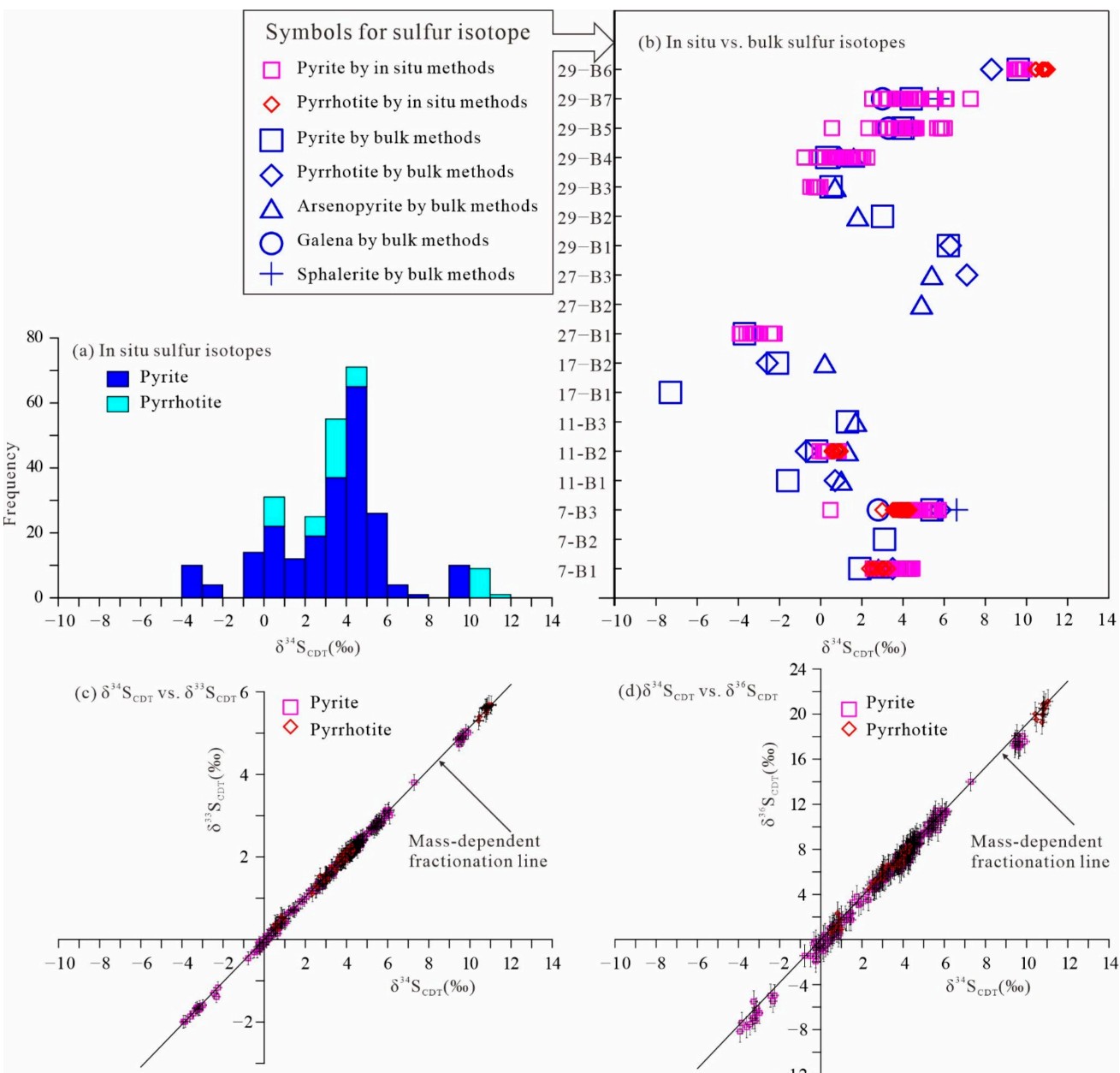

**Figure 10.** In situ sulfide isotopes of sulfides in ores from the Shenshuitan deposit. (**a**) Histogram of in situ sulfur isotopes; (**b**) diagram of in situ vs. bulk sulfur isotopes for each sample ("XI-3390NCM" at the beginning of each sample number has been omitted); (**c**) $\delta^{34}S_{CDT}$ vs. $\delta^{33}S_{CDT}$ plot; (**d**) $\delta^{34}S_{CDT}$ vs. $\delta^{34}S_{CDT}$ plot.

In addition, there was no apparent mass-independent fractionation of sulfur isotopes for sulfides in the Shenshuitan gold deposit because of near-zero $\Delta^{33}S$ values ranging from $-0.19‰$ to $0.15‰$ and near-zero $\Delta^{36}S$ values ranging from $-1.47‰$ to $+0.77‰$ (Supplementary Materials Table S4, Figure 10c,d).

### 4.4. Lead Isotope Results

A total of 41 sulfide minerals, separated from 18 ore samples from the Shenshuitan gold deposit, had relatively variable Pb isotope compositions, with $^{206}Pb/^{204}Pb$ ratios of

18.071 to 19.341, $^{207}$Pb/$^{204}$Pb ratios of 15.530 to 15.673, and $^{208}$Pb/$^{204}$Pb ratios of 37.908 to 38.702 (Supplementary Materials Table S5, Figure 11a,b). Different sulfide separates had relatively similar Pb isotope compositions (Figure 11c,d). In addition, different ore subtypes also had relatively similar Pb isotope compositions (Figure 11e,f). Sulfide separates from ore subtype 1 had highly variable Pb isotope ratios, with $^{206}$Pb/$^{204}$Pb ratios ranging from 18.071 to 18.341, $^{207}$Pb/$^{204}$Pb ratios from 15.530 to 15.658, and $^{208}$Pb/$^{204}$Pb ratios from 37.908 to 38.702. Sulfide separates from ore subtype 2 had relatively invariable Pb isotope ratios, with $^{206}$Pb/$^{204}$Pb ratios ranging from 18.345 to 18.521, $^{207}$Pb/$^{204}$Pb ratios from 15.565 to 15.673, and $^{208}$Pb/$^{204}$Pb ratios from 38.295 to 38.667. Sulfide separates from ore subtype 3 had relatively invariable Pb isotope ratios, with $^{206}$Pb/$^{204}$Pb ratios ranging from 18.418 to 18.454, $^{207}$Pb/$^{204}$Pb ratios from 15.627 to 15.666, and $^{208}$Pb/$^{204}$Pb ratios from 38.449 to 38.576. As with sulfur isotopes, ore subtype 1 had highly variable Pb isotope ratios compared to others mostly because it was the most complicated mixture of ore-forming fluid and host rocks.

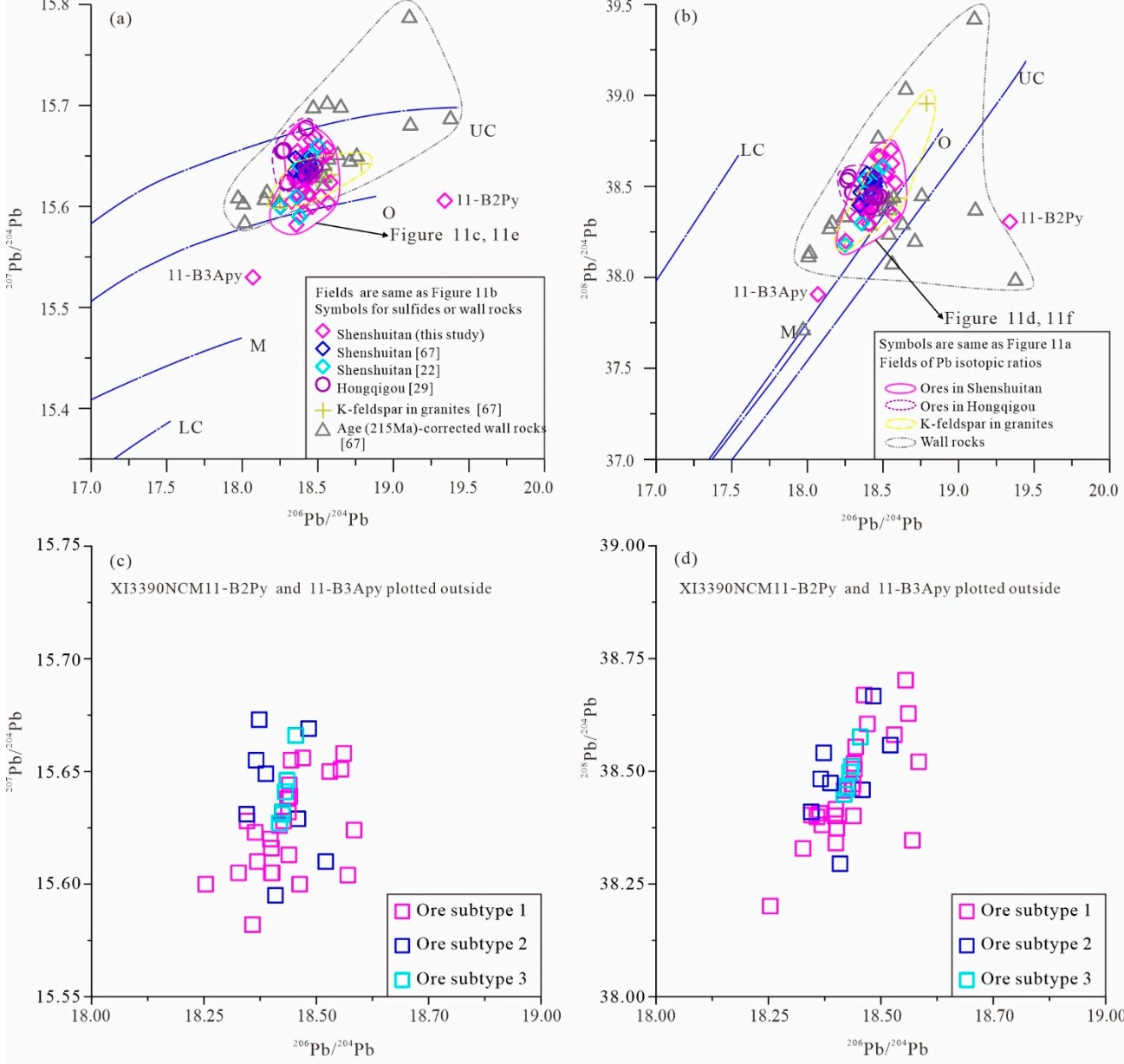

**Figure 11.** *Cont.*

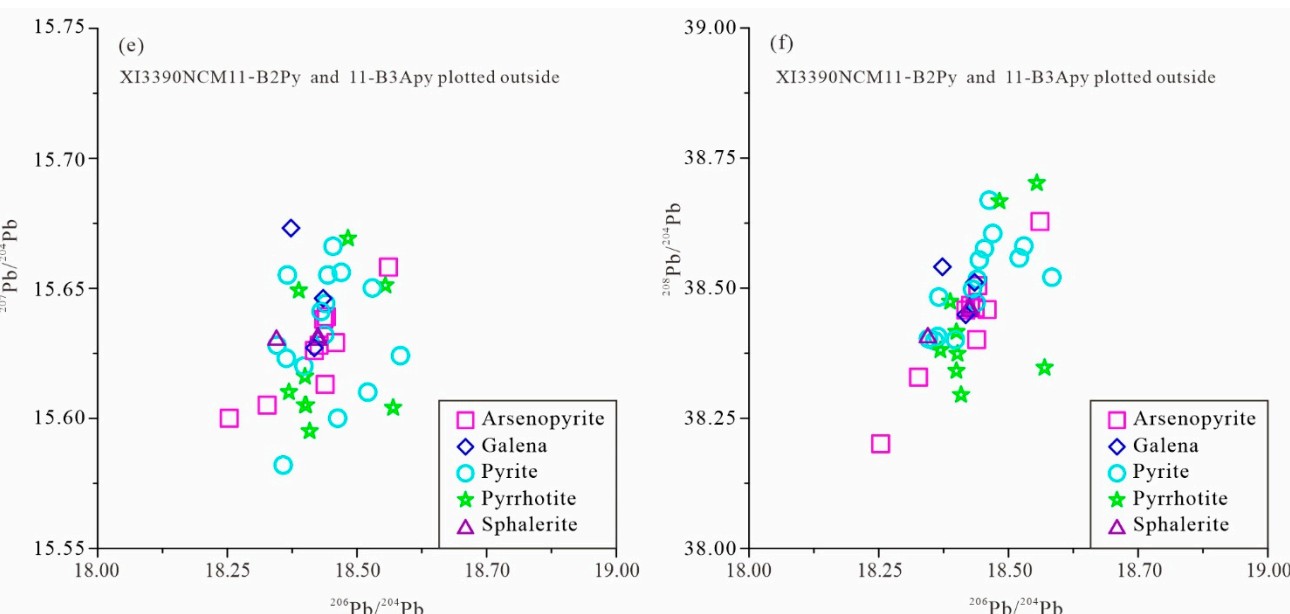

**Figure 11.** Lead isotope compositions of sulfides, K-feldspars, and wall-rocks in the Shenshuitan gold deposit. (**a**) Diagram of $^{207}Pb/^{204}Pb$ vs. $^{206}Pb/^{204}Pb$; (**b**) diagram of $^{208}Pb/^{204}Pb$ vs. $^{206}Pb/^{204}Pb$; (**c**) partially enlarged diagram of $^{207}Pb/^{204}Pb$ vs. $^{206}Pb/^{204}Pb$ for different ore subtypes in the Shenshuitan gold deposit; (**d**) partially enlarged diagram of $^{208}Pb/^{204}Pb$ vs. $^{206}Pb/^{204}Pb$ for different ore subtypes in the Shenshuitan gold deposit; (**e**) partially enlarged diagram of $^{207}Pb/^{204}Pb$ vs. $^{206}Pb/^{204}Pb$ for different minerals in the Shenshuitan gold deposit; (**f**) partially enlarged diagram of $^{208}Pb/^{204}Pb$ vs. $^{206}Pb/^{204}Pb$ for different minerals in the Shenshuitan gold deposit. UC, upper crust; O, orogen; M, mantle; LC, lower crust. The average growth curve is from [68]. Previously published data are from [22,29,67].

## 5. Discussion

### 5.1. Origins of Ore-Forming Fluid

Although the geological characteristics of the Shenshuitan gold deposit show similar features to orogenic gold deposits defined by [30,31], the $\delta^{18}O_{VSMOW}$ values (4.6‰ to 12.0‰) of the quartz and sericite within phyllic rock samples at the Shenshuitan deposit are different from those of quartz veins from Archean to Cenozoic orogenic lode gold deposits elsewhere ($\delta^{18}O$ = 12‰ to 22‰, [69]). Using 262 °C as the trapping temperature, the calculated $\delta^{18}O_{H2O}$ values of the fluids based on the average homogenization temperature show a variation from −3.8‰ to 3.6‰ with an average of 1.5‰ (Supplementary Materials Table S2), relatively comparable to $\delta^{18}O_{H2O}$ values of 1.7‰ to 6.7‰ previously calculated with 262 °C and $\delta^{18}O_{quartz}$ data published by other researchers [9,22,66]. Generally, trapping temperatures are most likely higher than the homogenization temperatures for the ore-forming fluids. Therefore, the $\delta^{18}O_{H2O}$ values of the fluids in the Shenshuitan gold deposit would be slightly higher than the values of −3.8‰ to 3.6‰ calculated for the average homogenization temperature of 262 °C or even the values of −0.6‰ to 6.8‰ calculated for the highest homogenization temperature of 355 °C. The $\delta^{18}O_{H2O}$ values for fluids of magmatic origin are mostly less than 9‰, whereas metamorphic origins usually have values higher than 3‰ [65]. In Figure 8, the $\delta^{18}O_{H2O}$ values of most quartz veins calculated for 355 °C in the Hongqigou gold deposit are apparently higher than 9‰, more likely indicating a metamorphic origin (Figure 8). Consequently, we infer that the original ore-forming fluid in the Shenshuitan gold deposit might have been metamorphic water similar to that in the Hongqigou gold deposit. Such metamorphic water migrated through deep regional faults upward into upper subsidiary ductile faults, e.g., fault XI, and strong water–rock reaction occurred between the metamorphic water and host rocks of fault breccia, granite, and slate. Then, phyllic alteration and corresponding mineralization happened

along the ductile faults and in host rocks in the Shenshuitan gold deposit. As surrounding meteoric water could easily flow into these upper subsidiary ductile faults, the original metamorphic water may have been mixed with some meteoric water during phyllic alterations and corresponding mineralization; a magmatic origin should be excluded because the ore-forming fluid is of low salinity, and no daughter minerals are trapped in the fluid inclusions [22].

Fluid inclusions in five quartz samples from the Shenshuitan gold deposit have $\delta D$ values varying from $-113.9‰$ to $-103.6‰$ (Supplementary Materials Table S2), which fall within the lower range of published data ($\delta D = -121.5‰$ to $-63.0‰$ [9,22]), and partially plot in the field of devolatilization of organic matter in sediments (Figure 8). Gray et al. [70] reported relatively similar $\delta D$ values in the range of $-70‰$ to $-100‰$ for fluid inclusions from quartz in some turbidite-hosted gold deposits in the Victorian gold province, Eastern Australia, and argued that the origin of the fluids was the surrounding metasedimentary rocks [70]. The low $\delta D$ values obtained in some lode gold deposits have been interpreted to result from the reaction between deep-sourced non-meteoric fluids and $\delta D$-depleted organic matter in host rocks [70–73]. However, there are no reports of organic matter in the sedimentary wall rock of the Shenshuitan deposit. The origin from $\delta D$-depleted organic matter should be excluded. In some cases, low $\delta D$ values from bulk extraction of fluid inclusions are often considered to reflect secondary inclusions formed in the presence of meteoric water during the uplift of the deposits [63,72,74,75]. The $\delta D$ values of quartz in the Shenshuitan gold deposit are significantly lower than the $= \delta^{18}D_{H2O}$ values of the fluids calculated on the basis of the average homogenization temperature from four sericite samples in this study and published data ($\delta D = -74.5‰$ to $-68.3‰$ [66]; Figure 8). Thus, we believe that secondary inclusions formed in the presence of meteoric water during the uplift of the deposits contribute to the strong depletion of H isotope compositions from the bulk extraction of fluid inclusions within quartz. The explanation of low $\delta D$ values resulting from the presence of $H_2$ and/or $CH_4$ in the aqueous fluid [76] can also be excluded because there are only trace amounts of $H_2$ and/or $CH_4$ in the aqueous fluid according to Raman spectroscopy [25]. Thus, higher calculated $\delta D$ values of sericite ranging from $-74.5‰$ to $-68.3‰$ may be closer to the $\delta D$ values of original metamorphic ore-forming fluids.

Furthermore, the H and O isotopes of ore-forming fluids in the Hongqigou gold deposit are closer to metamorphic water, as shown in Figure 8. It has been considered that most quartz vein ores yield enriched $\delta^{18}O_{H2O}$ and more invariable $\delta D$ values than most phyllic rock ores, which perhaps indicates that there is no apparent mixing of meteoric water during the mineralization of a quartz vein, but there is an obvious introduction of meteoric water during the mineralization of phyllic rock ores. It is thus inferred that the original metamorphic ore-forming fluids in the Shenshuitan gold deposit may have been mixed with meteoric water during phyllic alterations and corresponding gold mineralization.

### 5.2. Sulfur Sources

The $\delta^{34}S$ values of sulfide minerals from gold-bearing ores of the Shenshuitan gold deposit vary extensively from $-7.3‰$ to $9.6‰$, with an average of $2.5‰$ (Figure 9a), which span the range of data previously published by other researchers from $0.4‰$ to $8.5‰$ [9,67]. Our sulfur isotope compositions also span the reported $\delta^{34}S$ values for most orogenic gold deposits elsewhere (typically $\delta^{34}S = 0–9‰$ [30,31,72,77,78]). As we know, sulfur isotope signatures are extremely variable for orogenic gold deposits and reported $\delta^{34}S$ values have been shown to be as low as $-20‰$ and as high as $25‰$ for sulfide minerals from orogenic gold deposits [31]. Therefore, our highly variable $\delta^{34}S$ values are reasonable, and they are broadly similar to some of the sediment-hosted orogenic gold deposits (e.g., $-6.3‰$ to $9.3‰$, [79]). However, the Shenshuitan gold deposit was hosted in the contact between granite and slate, and it should not be simply classified into a sediment-hosted orogenic gold deposit.

The presence of pyrrhotite + pyrite + arsenopyrite, and the alteration assemblage of quartz + sericite in the Shenshuitan gold deposit, together with the lack of oxidized phases

and sulfate minerals, indicate that sulfur was present in the hydrothermal fluids mainly as reduced sulfur ($H_2S$) [80,81]. The $\delta^{34}S_{VCDT}$ values for $H_2S$ ($\delta^{34}S_{H2S}$) in equilibrium with sulfides were also estimated by evaluating the $\delta^{34}S_{VCDT}$ values of pyrite, pyrrhotite, galena, and sphalerite using the equilibrium isotopic fractionation factor of sulfide with respect to $H_2S$ suggested by [81], as well as the inferred average temperature of the hydrothermal fluid during the sulfide mineralization event of 262 °C. The calculated $\delta^{34}S_{H2S}$ values for $H_2S$ in the hydrothermal fluids range from −8.7‰ to 8.2‰, with an average value of 1.9‰ (Supplementary Materials Table S3, Figure 9g). The near-zero average $\delta^{34}S_{H2S}$ of the fluids computed in our study may suggest a possible derivation from sources with average crustal sulfur composition [82,83]. However, the large range (16.9‰) of $\delta^{34}S_{sulfide}$ or $\delta^{34}S_{H2S}$ indicates a mixing source of sulfur. In particular, on the basis of the large difference between the ore subtype 1 in the granite and ore subtype 2 in the slate (Figure 9b,d), we infer that the sulfur may have been mixed from a unique original ore-forming fluid and different host rocks (such as brecciated granite or slate). Low-grade metamorphism can release significant Au, S, and other elements from the source rocks to be subsequently concentrated by hydrothermal processes, forming orogenic gold deposits at higher crustal levels [84]. The sulfur isotope composition of the resulting S-bearing fluid in the Shenshuitan gold deposit may be similar to that of sulfides from quartz veins in the Hongqigou gold deposit which exhibited the heaviest $\delta^{34}S$ values (Figure 9e, [29]). It is inferred that most quartz veins such as those in the Hongqigou gold deposit yield heavier $\delta^{34}S$ than most phyllic rock ores such as those in the Shenshuitan gold deposit, indicating that there is no apparent reaction between the ore-forming fluid and host rock during the mineralization of a quartz vein. However, obvious water–rock reaction during mineralization and alteration of phyllic rock ores must happen, accompanied by replacement and mixing of S isotopes between the original metamorphic S-bearing fluid and different host rocks. Perhaps the many introduced fault breccias and fault gouges within the faulted zone contribute the largest variation to sulfur isotopes within the contact of granite and slate, as sulfides in ore subtype 1 have the largest variation in sulfur composition and corresponding $\delta^{34}S_{H2S}$ values, as shown in Figure 9g. Furthermore, the sulfides in ore subtype 2 have relatively heavy $\delta^{34}S$ values, partially because of possibly higher original $\delta^{34}S$ values of the slate (unfortunately, we did not analyze any barren slate), and partially because of the highest $\delta^{34}S$ values of the original ore-forming fluids. The similarly heavy $\delta^{34}S$ values of the sulfides in ore subtypes 3 are apparently due to the weak interaction between water and phyllic rock of ore subtype 1 while quartz + calcite + sulfide intruded ore subtype 1, and the $\delta^{34}S$ values were likely inherited mostly from the heaviest sulfur isotope of the ore-forming water as was the case in the Hongqigou deposit.

In summary, the original sulfur source of the ore-forming fluids of the Shenshuitan gold deposit may have the heaviest $\delta^{34}S$ values similar to the Hongqigou gold deposit, and the different ore subtypes have variable S isotope compositions that may account for different degrees of water–rock reaction/mixing and different host rock types. Therefore, sulfur sources for the sulfides in the Shenshuitan gold deposit include those from original metamorphic ore-forming fluids with the heaviest $\delta^{34}S$ values and those from granite or slate host rocks with variable $\delta^{34}S$ values.

Because in situ sulfur compositions are comparable to those in the bulk samples (Figure 10b), the same conclusion for the sulfur source of the Shenshuitan gold deposit can be obtained from the in situ samples. In addition, because there is no apparent mass independent fractionation of sulfur isotopes for sulfides in the Shenshuitan gold deposit (Figure 10c,d), the mass-independent fractionations of sulfur ($\Delta^{33}S$ and $\Delta^{36}S$) cannot be used as tracers to characterize gold mineralizations in the Shenshuitan gold deposit. This is mostly because samples younger than 2.45 Ga are interpreted as carrying a mass conservation-type signal associated with mass-dependent fractionation (i.e., [62,85]).

### 5.3. Lead Sources

Sulfide minerals usually contain very low concentrations of U and Th, as well as insignificant radiogenic Pb isotopes [86]. Therefore, Pb isotopes of sulfides from ore deposits are commonly used to constrain the source of Pb regardless of the total Pb concentration of sulfides (e.g., [64,87–91]). In this study, most Pb isotope compositions for studied sulfides in the Shenshuitan gold deposit clustered as a group (pink solid line in Figure 11a,b) close to the orogen lead reservoir [68]; the few abnormal Pb isotope compositions (the different Pb compositions as displayed by samples 11-B2Py and 11-B3Apy) can be ignored. Meanwhile, most previously published Pb isotope compositions [22,67] also plotted in the field of our sulfide group (Figure 11). The Pb isotope compositions of sulfides in the Hongqigou gold deposit [29], those of potassic feldspars in granite wall rocks, and age (215 Ma)-corrected ones of bulk rock of sedimentary wall rocks [67] are also shown in Figure 11a,b. A corrected age of 215 Ma can be used for the original Pb isotope composition calculation because 215 Ma most likely approximates the mineralized age for the Shenshuitan gold deposit [4].

The field for the Pb isotope composition of sulfides in the Shenshuitan gold deposit overlapped partly with that of the Hongqigou gold deposit (purple dashed line in Figure 11a,b) or potassic feldspars in the granite host rock (yellow dashed line in Figure 11a,b). The sulfides in quartz veins from the Hongqigou gold deposit yielded relatively invariable Pb isotope compositions, which is consistent with no apparent mixing between ore-forming fluids and host rocks. Given that the ore-forming fluids in the Shenshuitan gold deposit are the same as those in the Hongqigou gold deposit as mentioned above, the Pb isotope compositions of sulfides in the Shenshuitan gold deposit would be similar to each other if no replacement happened during mineralization. However, phyllic rock ores, which are products of strong water–rock reaction between ore-forming fluids and host rocks, are actually dominant in the Shenshuitan gold deposit. Furthermore, the Pb isotope compositions of sulfides in the Shenshuitan gold deposit were roughly located between those of the Hongqigou deposit and those of the potassic feldspars in the granite wall rock (Figure 11a,b), perhaps indicating that the former was a mixture between ore-forming fluids in the Shenshuitan gold deposit and granites. Moreover, the above three Pb isotope compositions mostly plotted within the age (215 Ma)-corrected sedimentary wall rocks in the uranogenic plot (gray dashed line in Figure 11a) and totally plotted within it in the thorogenic plot (gray dashed line in Figure 11b), if a few abnormal Pb isotope compositions falling outside the plot can be ignored. Therefore, host rocks of silicic slate also contribute to lead sources for sulfides in the Shenshuitan gold deposit. Strong water–rock reaction resulted in phyllic alteration and mineralization in ductile fault zones and near host rocks, as well as mixing of Pb isotopes.

Consequently, the Pb isotope compositions of different sulfide minerals in the Shenshuitan gold deposit could be interpreted to reflect a mixture between lead from the original ore-forming metamorphic fluid similar to that in the Hongqigou gold deposit and lead within host rocks of the granite or silicic slate. This argument was similar to that presented on the basis of sulfur isotope research in the Shenshuitan gold deposit. Moreover, the lead isotope compositions for these sulfides obviously spanned the orogenic line [68] (Figure 11a,b), indicating some affinities to an orogenic lead reservoir, which is consistent with the tectonic evolutionary history of the EKO (e.g., [4]).

### 5.4. The Genetic Type and Possible Metallogenic Model

Based on the results described above and a comparison with typical orogenic gold deposits, we propose that the Shenshuitan gold deposit belongs to the orogenic gold category, as described by [30,92]. Many of the geological and geochemical features exhibited by this deposit are consistent with those of orogenic gold deposit, including the following:

(1)    the tectonic location of the deposit located within the EKO, a Late Triassic orogenic belt;
(2)    the gold mineralization hosted by the contact between the phyllic brecciated granite and pyritized slate, and controlled by ductile faults within the contact;

(3)   the major sulfide minerals represented by pyrite and arsenopyrite, with minor chalcopyrite, galena, sphalerite, and pyrrhotite;

(4)   the alteration assemblage including silicification and sericitization, with minor carbonatization and chloritization;

(5)   the major ore-forming fluids showing $CO_2$-bearing fluids with low salinity (0.21–17.36 equiv. wt% NaCl), and low–intermediate homogenization temperatures from 141 to 355 °C [22];

(6)   the estimated $\delta D$ values ($-107$‰ to $-63$‰) and $\delta^{18}O$ values ($-4.6$‰ to 11.3‰) for fluid, indicating a mixture of metamorphic water and meteoric water;

(7)   the bulk $\delta^{34}S$ values of sulfides ranging from $-7.3$‰ to 9.6‰, as well as in situ $\delta^{34}S$ values of sulfides ranging from $-3.92$‰ to 11.04‰, covering values of typical orogenic gold deposits (0–9‰ [30,31,72,77,78]);

(8)   the bulk Pb isotope compositions of sulfides, with $^{206}Pb/^{204}Pb$ ratios of 18.071–9.341, $^{207}Pb/^{204}Pb$ ratios of 15.530–15.673, and $^{208}Pb/^{204}Pb$ ratios of 37.908–38.702, indicating some affinities to an orogenic lead reservoir [68], which is consistent with the tectonic evolutionary history of the EKO.

We argued that the gold mineralization was strongly controlled by fault XI with phyllic alteration, which cut the 215 Ma Huanglonggou quartz diorite dyke in the Shenshuitan gold deposit [28]. Until a more accurate direct age for such low-sulfide Au mineralization can be obtained, we believe that 215 Ma (the Late Triassic) may be a good approximate mineralized age for the Shenshuitan gold deposit. As stated by [30], orogenic gold deposits usually form during compressional to transpressional deformation processes at convergent plate margins in accretionary or collisional orogens. In general, the metamorphic model for orogenic gold formation is commonly accepted, in which ore-forming metallogenic fluids are usually produced at depth during the prograde greenschist-to-amphibolite facies metamorphism of supracrustal rocks [31,92]. However, there are some important Phanerozoic orogenic gold deposits, e.g., the Cretaceous orogenic gold deposits on the margins of the eastern North China block, which clearly do not fit this model, because the Precambrian crustal rocks that host these orogenic gold deposits experienced high-grade metamorphism long before ore formation [31]. Similarly, ancient metamorphic fluid, perhaps produced during the Precambrian metamorphism of gneiss, schist and phyllite in the Wulonggou gold field, cannot also explain the Late Triassic orogenic gold mineralization found in this study. The collision between the EKO and the BH–SG was considered soft in the Late Triassic [93], which suggested that extensive folding, thrusting, and faulting by continued compression [34,94] occurred in the EKO during the final closure of the Paleo-Tethys at the end of the Late Triassic rather than clear syn-collisional magmatism [4]. In this case, collision-related thermal events, episodically raising geothermal gradients within the deeply buried metamorphic units, possibly initiated some metamorphic fluids and drove long-distance metamorphic fluid migration along such deep-seated, steeply dipping regional ductile faults or shear zones and their subsidiary faults. However, it is more likely that the soft collision between EKO and BH–SG cannot produce metamorphism among deep rocks and subsequent metamorphic fluids at depth. If this is the case, the slab devolatilization model proposed for the Jiaodong gold province [92] may also be applicable to the Shenshuitan gold deposit, as well as the Wulonggou gold field. In their model, fluids are being added directly from the late-orogenic metamorphic devolatilization of stalled subduction slabs and oceanic sediments, and the overpressured slab-derived metamorphic fluids can also be channeled directly into crustal-scale faults with subsequent upward flow [34, 92]. Regardless of the model used to generate deep metamorphic fluids, different scales of ductile faults exert important structural control on orogenic gold mineralizations in the EKO, as exemplified by the Shenshuitan gold deposit. The first-order crustal-scale fault might have served as the major channel for upward migration of the deeply generated metamorphic fluids (Figure 12). These original deep metamorphic fluids may have migrated through the regional faults upward into the subsidiary ductile faults at higher levels, and have remobilized gold, sulfur, and lead from the surrounding metalliferous

metasedimentary rocks. During the migration of these metamorphic fluids, sulfur, lead, and gold were leached from the host rocks; eventually, the gold-bearing phyllic rocks were formed within subsidiary ductile faults along the contact of granite and slate at higher crustal levels.

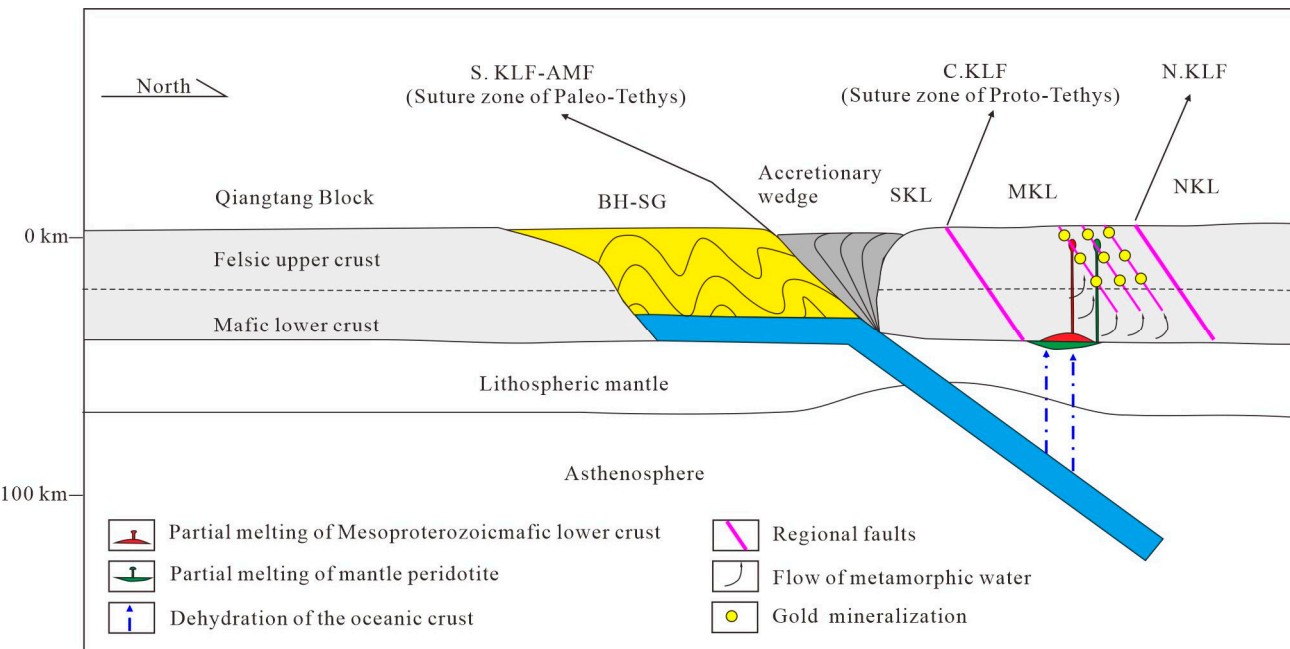

**Figure 12.** Metallogenic model for the Shenshuitan gold deposit, as well as gold deposits within the Wulonggou gold field in the EKO. The corresponding Late Triassic collision model between the BH–SG and EKO is modified after [10]. All abbreviations are the same as in Figure 1.

Considering our data, the significance of this study for further exploration in the study area and perhaps in the EKO can be inferred. Firstly, the relatively extensional space with ductile faults should be an important ore-forming location, because it facilitates filling of the original metamorphic water and subsequent replacement and mixing of water and host rocks. Secondly, higher $\delta D$, $\delta^{18}O$, $\delta^{34}S$, and Pb isotopes for gold deposits in the Shenshuitan gold deposit are perhaps a better indication of originating or being sourced from the original metamorphic water, as well as likely greater gold mineralization potential.

## 6. Conclusions

The following conclusions were drawn from the interpretation of the H, O, S, and Pb isotope compositions of ores from the Shenshuitan gold deposit in the EKO reported in this study:

(1) The measured $\delta^{18}O$ values of quartz and sericite from Phyllis rock ores in the Shenshuitan gold deposit ranged from 4.6‰ to 12.0‰, and the estimated $\delta^{18}O$ values of fluids ranged from $-3.8$‰ to 3.6‰, calculated for the average homogenization temperature of 262 °C, or from $-0.6$‰ to 6.8‰, calculated for the highest homogenization temperature of 355 °C. The $\delta D$ values of fluid inclusions in gold-bearing quartz or estimated $\delta D$ values from sericite ranged from $-113.9$‰ to $-73.1$‰. The results suggest that the original metamorphic ore-forming fluids in the Shenshuitan gold deposit might have been metamorphic water similar to that in the Hongqigou gold deposit, which may have been mixed with meteoric water during phyllic alterations and corresponding gold mineralizations.

(2) Bulk $\delta^{34}S$ values of gold-associated hydrothermal sulfides in the Shenshuitan gold deposit ranged from $-7.3$‰ to 9.6‰, and in situ $\delta^{34}S$ values ranged from $-3.92$‰ to 11.04‰, located within the extremely variable sulfur isotope range as low as

−20‰ and as high as 25‰ for sulfide minerals from orogenic gold deposits [31]. The sulfur isotope composition of the original S-bearing metamorphic fluid in the Shenshuitan gold deposit may be similar to that of the Hongqigou gold deposit, suggesting that the sulfur sources of sulfides in the Shenshuitan gold deposit included those of original metamorphic ore-forming fluids with the heaviest $\delta^{34}$S values and others from host rocks of granite or slate with variable $\delta^{34}$S values. Because obvious water–rock reaction should happen during phyllic alteration and mineralization, these interations must have been accompanied by the mixing of S isotopes between the original metamorphic S-bearing fluid and different host rocks.

(3) Lead isotope compositions of sulfides from gold-associated hydrothermal sulfides in the Shenshuitan gold deposit ranged from 18.071 to 19.341 for $^{206}$Pb/$^{204}$Pb ratios, from 15.530 to 15.673 for $^{207}$Pb/$^{204}$Pb ratios, and from 37.908 to 38.702 for $^{208}$Pb/$^{204}$Pb ratios. These Pb isotope compositions of sulfides overlapped partly with those of the Hongqigou gold deposit or the granite host rock, and mostly lay within the values of the age (215 Ma)-corrected sedimentary wall rocks. We interpreted these data to reflect a mixed source for Pb, involving the original ore-forming metamorphic fluid and lead sources within host rocks of the granite or silicic slate. The lead isotope compositions for these sulfides obviously spanned the orogenic line, indicating some affinities to an orogenic lead reservoir, which is consistent with the tectonic evolutionary history of the EKO.

(4) The Shenshuitan gold deposit can be classified as an orogenic gold deposit. Following deformation associated with the final soft collision between BH–SG and EKO in the Late Triassic, i.e., the final closure of the Paleo-Tethys Ocean, regional ductile faults, and subsequent ore-controlling subsidiary ductile faults were developed; original deep metamorphic ore-forming fluids were then channeled along these faults. During migration of fluids, sulfur, lead, and gold were leached from the host rocks and mixed with those in the original metamorphic water; eventually, the gold-bearing phyllic rocks were formed within subsidiary ductile faults via contact with granite and slate at higher crustal levels.

**Supplementary Materials:** The following are available online at https://www.mdpi.com/article/10.3390/min12030339/s1: Table S1. Sample list in the Shenshuitan gold deposit; Table S2. H, O isotope data of quartz or sericites from the Shenshuitan and Hongqigou gold deposits; Table S3. Bulk sulfur isotope data of sulfides from the Shenshuitan and Hongqigou gold deposits; Table S4. In situ multiple sulfur isotope data using SIMS for sulfides from the Shenshuitan gold deposit; Table S5. Pb isotope compositions for sulfides from the Shenshuitan and Hongqigou gold deposits.

**Author Contributions:** Conceptualization, Q.-F.D.; methodology, L.C., K.S. and F.L.; software, X.Z., L.C., K.S., and F.L.; validation, X.Z., L.C., K.S. and F.L.; formal analysis, X.Z., L.C., K.S., F.L. and Q.-F.D.; investigation, X.Z., Q.-F.D., L.C. and F.L.; resources, T.P. and Q.-F.D.; data curation, X.Z. and Q.-F.D.; writing—original draft preparation, X.Z. and Q.-F.D.; writing—review and editing, Q.-F.D. and Y.G.; visualization, X.Z. and Q.-F.D.; supervision, T.P. and Q.-F.D.; project administration, Q.-F.D.; funding acquisition, T.P. and Q.-F.D. All authors read and agreed to the published version of the manuscript.

**Funding:** This research was funded by the National Natural Science Foundation of China, grant number 41572056, and the Special Funding for Qinghai Scholars, grant number QHS201802.

**Informed Consent Statement:** Informed consent was obtained from all subjects involved in the study.

**Data Availability Statement:** All data generated or used during the study appear in the submitted article.

**Acknowledgments:** We thank Marco Fiorentini, Crystal LaFlamme, and Dennis Sugiono from the UWA for their help with in situ sulfur isotope analyses, and we also acknowledge the facilities and technical assistance of the Australian Microscopy and Microanalysis Research Facility at the CMCA, UWA. We are also grateful to the Analytical Laboratory of BRIUG, CNNC for their help with bulk isotope analyses. We also thank Yu Han and Yuan-Liang Deng in the First Institute of the Qinghai Geology Survey for their assistance during the field work. Their constructive, stimulating, and valuable comments and suggestions helped improve the manuscript significantly. We thank LetPub (www.letpub.com, accessed on 14 January 2022) for its linguistic assistance during the preparation of this manuscript (14 January 2022).

**Conflicts of Interest:** The authors declare no conflict of interest.

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
