# Peer review of "Isotope Geochemistry of the Shenshuitan Gold Deposit within the Wulonggou Gold Field in the Eastern Kunlun Orogen, Northwest China: Implications for Metallogeny"

_minerals, doi:10.3390/min12030339_

Round 1

Reviewer 1 Report

Dear authors,

I read your paper, you have good results about one of gold deposits in EKO. Below is a summary of the my comments to be considered in the revision of the paper.

  • There are many (Error! Ref-108 erence source not found.) in text. Needs to be fixed.
  • There are many line errors in the text, like line 34, 58, 88, 156
  • Number of gold deposits in Figure 2 hardly seen. There are also many yellow spots on the map not mentioned in the legend .
  • There is a general problem of massive citation: putting lots of citations together after a general statement makes it impossible to know who said what.
  • Although there is a fair amount of literature review, most of them from earlier 2000, recent papers can be mentioned in the text and added to references.
  • Paragenetic study is very important for stable isotope analyses, but there is a lack of such a study. All what is shown is a few photos (Figure 5. The symbols of the minerals hardly seen, can you make them more visible? )Also a paragenetic sequence diagram, should be provided.
  • Information about ore forming fluids (fluid inclusion studies) is very poor, I agree FI studies mentioned from previous studies, but FI data is extremely important as used for calculated O&H isotope results. At least some pictures showing the actual occurrences and relationships of different types of fluid inclusions should be provided and discussed. I also not sure if reference (22) given for previosu FI study was correct? should be (21) or?
  • The effect of temperature on calculated O isotopes of fluids is very large, therefore, it is important to clearly specify the temperature used in the calculation, both in the text and in the caption of the figure. In addition, I would suggest you to use the range of temperature, rather than the average, in the calculation. This will give a more objective evaluation of the meaning of the calculated O isotope data, with uncertainties being clearly shown.
  • There are many dublications in the discussion. This section could be improved to better reflect the large amount of information reviewed in relation to the title/objective of the paper. In my view, the conclusions also should be expanded to better summarize the overall "feel" of the main review section to give the reader a strong take home message.
  • Finally, since you have mentioned the importance of the studies like this one are important for mineral exploration, I would expect to see some discussion about the significance of this study for further exploration in the study area and perhaps in the Eastern Kunlun Orogen (EKO).
  • Best regards

Author Response

Dear Reviewer 1,

Many thanks for your comments on our manuscript entitled "Isotope Geochemistry of the Shenshuitan Gold Deposit within the Wulonggou Gold Field in the Eastern Kunlun Orogen, Northwest China: Implications for Metallogeny". We have revised the manuscript according to your comments. And we have submitted our revised manuscript to English editing services listed at https://www.mdpi.com/authors/english for a *professional English editing*, as you have pointed out that the *English language* of our manuscript does not meet necessary standards and requirements. Below are our detailed response to your comments:

  1. There are many (Error! Ref-108 erence source not found.) in text. Needs to be fixed.

Response: “Error! Reference source not found.” are replaced as “Figure”. All figure hyperlinks of cross-reference in the original manuscript showed such errors. We think that these errors are automatically caused by the review system or a different Word version.  

  1. There are many line errors in the text, like line 34, 58, 88, 156

Response: All line errors like “22]” in line 58 in the text have been fixed in the revised manuscript. Some citations which were inserted by “Endnote” in the original manuscript exhibited such errors. We think that these errors are automatically caused by the review system or a different Word version.  

  1. Number of gold deposits in Figure 2 hardly seen. There are also many yellow spots on the map not mentioned in the legend.

Response: We have modified the Figure 2. Number of gold deposits in Figure 2 were marked with black fonts. Except four major gold deposits, other gold deposits are small scale ones, so their names and numbers were omitted in Figure 2.

  1. There is a general problem of massive citation: putting lots of citations together after a general statement makes it impossible to know who said what.

Response: We have tried our best to revise these massive citations, except those of the Yanjingou gold deposit. The Yanjingou gold deposit have massive published articles, so the massive citations have no apparent troubles.

  1. Although there is a fair amount of literature review, most of them from earlier 2000, recent papers can be mentioned in the text and added to references.

Response: Thanks for your suggestions. We have added some recent papers cited in the revised manuscript, such as Goldfarb et al. (2015) and Groves et al. (2016). There are other 32 papers from 2011 to 2020 have been already cited in the original manuscript.

  1. Paragenetic study is very important for stable isotope analyses, but there is a lack of such a study. All what is shown is a few photos (Figure 5. The symbols of the minerals hardly seen, can you make them more visible? )Also a paragenetic sequence diagram, should be provided.

Response: We have provided the new Figure 5 with larger purple fonts for symbols of the minerals and some high resolution photomicrographs. We have also achieved more detailed paragenetic study in the revised manuscript and provided Figure 6 showing paragenetic sequences of minerals in different ore-forming stages.

  1. Information about ore forming fluids (fluid inclusion studies) is very poor, I agree FI studies mentioned from previous studies, but FI data is extremely important as used for calculated O&H isotope results. At least some pictures showing the actual occurrences and relationships of different types of fluid inclusions should be provided and discussed. I also not sure if reference (22) given for previosu FI study was correct? should be (21) or?

Response: We are terribly sorry that we can not supply detailed fluid inclusion studies in this study, because we have no enough thin sections for fluid inclusion studies. Reference 22 of Zhang (2018) have completed certain fluid inclusion studies and obtain 137 sets of FI data regarding ore subtype 1 and 2, so the homogenization temperatures ranging from 141 to 355°C can represent the FI data for the quartz-sulfide epoch (ore subtype 1 and 2) in the Shenshuitan gold deposit. Some FI pictures can be found in the appendix of photomicrographs in Zhang (2018).

  1. The effect of temperature on calculated O isotopes of fluids is very large, therefore, it is important to clearly specify the temperature used in the calculation, both in the text and in the caption of the figure. In addition, I would suggest you to use the range of temperature, rather than the average, in the calculation. This will give a more objective evaluation of the meaning of the calculated O isotope data, with uncertainties being clearly shown.

Response:The homogenization temperatures of quartz in ores of ore subtype 1 and 2 range from 141°C to 355°C, with an average of 262°C in the Shenshuitan gold deposit (Zhang ,2018). Such a wide range of temperatures is inconsistent with the lack of significant metal zoning, so the low-temperature homogenization measurements are mostly from post-ore inclusions and/or inclusions that have undergone post-entrapment modifification (e.g., Goldfarb et al., 2015). Using 262°C as the trapping temperature, the δ18OH2O values of the fluids (−3.8‰ to 3.6‰) for fluid inclusions from quartz samples were calculated and are shown in Table S2. Given the trapping temperature was defined by the highest homogenization temperature of 355°C, the calculated δ18OH2O values of the fluids have only a 3.2‰ increment and vary from −0.6 to +6.8‰ (Figure 8). Generally, trapping temperatures are most likely higher than the homogenization temperatures for the ore-forming fluids. Therefore, the δ18OH2O values of the fluids in the Shenshuitan gold deposit would be slightly higher than the values of −3.8‰ to 3.6‰ calculated for the average homogenization temperature of 262°C or even the values of −0.6‰ to 6.8‰ calculated for the highest homogenization temperature of 355°C. In Figure 8, δ18OH2O values of most quartz vein calculated for 355°C in the Hongqigou gold deposit are apparently have higher than 9‰, more likely indicating a metamorphic origin (Figure 8).

  1. There are many dublications in the discussion. This section could be improved to better reflect the large amount of information reviewed in relation to the title/objective of the paper. In my view, the conclusions also should be expanded to better summarize the overall "feel" of the main review section to give the reader a strong take home message.

Response:Thanks for your suggestions. We have deleted all duplications in the Discussion, and improved the text in the Discussion.We also expanded the Conclusions as your suggestion.

  1. Finally, since you have mentioned the importance of the studies like this one are important for mineral exploration, I would expect to see some discussion about the significance of this study for further exploration in the study area and perhaps in the Eastern Kunlun Orogen (EKO).

Response: Thanks for your suggestions. We have completed some discussion about the significance of this study for further exploration in the study area and perhaps in the EKO as follows: Considering our data, the significance of this study for further exploration in the study area and perhaps in the EKO can be inferred. Firstly, the relatively extensional space with ductile faults should be an important ore-forming location, because it facilitates filling of the original metamorphic water and subsequent replacement and mixing of water and host rocks. Secondly, higher δD, δ18O, δ34S, and Pb isotopes for gold deposits in the Shenshuitan gold deposit are perhaps a better indication of originating or being sourced from the original metamorphic water, as well as likely greater gold mineralization potential.

Kind Regards,

Qing-feng Ding

Reviewer 2 Report

  1. Line 9. Is there any specific feature that distinguishes this fault from others?
  2. Lines 12-15. Some key characteristics of host rocks must be provided here to support their contribution to ore-forming fluids.
  3. Lines 48-49. Here and multiple times elsewhere in the text, there is some bad wording. For instance, here, “genetic type…” cannot “belong to… deposit…”. It is the deposit that may belong to a certain genetic type, not opposite.
  4. Lines 56-64. Given the previous studies, including a recent one, which considers an adjacent Hongqigou deposit of similar type, is there any conundrum and/or scientific significance in defining the orogenic type of gold occurrence in this study (as emphasized in the abstract)? I believe not. Besides, in the Introduction, the authors would benefit from providing some key features, which allowed to classify the mentioned deposits as orogenic.
  5. Lines 302-315. I do not think there is a need to list all ranges of sulfur isotope composition for all sulfides in the text, given that there are several figures and a supplementary table with the original data. Instead, I would focus on highlighting the main similarities and differences, which are linked (or not) to types of mineral assemblages, their temperatures of precipitation etc.
  6. Line 347 and below. Same as for S – you don’t just list the Pb/Pb ranges, instead show as straightforward as possible there is (or is not) a dependence between Pb isotopes and mineralogy of rocks/ores, including possible links to host rocks or any other isotopic sources of Pb, which are used in figures for interpretation.
  7. Line 465 (and again – multiple times elsewhere in the text). I really doubt of calculating the average values in the ranges like these (e.g., from -9 to +8). This would not be a representative value in any case. For S, here I do not see the compromising explanation of S isotope variations based on mineralogical characteristics and tectonic history of the deposit. Why would the original fluid have the heaviest S?
  8. Line 522 and below. Again, for Pb isotopes, a few ephemeral isotopic reservoirs are listed, which might contribute to isotopic variations (which are pretty large), but no particular mechanisms were provided to account for mixing between them.
  9. Line 596 and further. You state that the fluid was oxidized (CO2-bearing), but above, the reduced S is suggested as a major form in the fluid (Line 460). It is better to constrain the oxygen fugacity ranges, which are relevant for an initial fluid and ore precipitation process. What do you think of a major barrier for precipitation? (cooling/decompression and boiling/fluid reduction etc.)?

Author Response

Dear Reviewer 2,

Many thanks for your comments on our manuscript entitled "Isotope Geochemistry of the Shenshuitan Gold Deposit within the Wulonggou Gold Field in the Eastern Kunlun Orogen, Northwest China: Implications for Metallogeny". We have revised the manuscript according to your comments. And we have submitted our revised manuscript to English editing services listed at https://www.mdpi.com/authors/english for a *professional English editing*, as you have pointed out that the *English language* of our manuscript does not meet necessary standards and requirements. Below are our detailed response to your comments:

  1. Line 9. Is there any specific feature that distinguishes this fault from others?

Response: No, there are not any specific feature that distinguishes fault XI from others. Three NW-trending ductile fault zones, which consist predominantly of fault I–fault XV, developed in the Wulonggou gold field. Qian et al. (1998), Qian et al. (1999), Li et al. (2001), and Lu (2011) argued such faults share similar features and host the Shenshuitan, Yanjingou, Hongqigou and Danshuigou gold deposits, as well as numerous gold occurrences throughout the entire Wulonggou gold field. We have simply described features of fault XI in the Section 2.2.

  1. Lines 12-15. Some key characteristics of host rocks must be provided here to support their contribution to ore-forming fluids.

Response: “Host rocks predominantly consist of Ordovician silicic slate and late Silurian granites” was inserted in the Abstract. Because there is a limit of 200 words maximum, some key characteristics of host rocks have been provided in the Section 2.2 to support their contribution to ore-forming fluids.

  1. Lines 48-49. Here and multiple times elsewhere in the text, there is some bad wording. For instance, here, “genetic type…” cannot “belong to… deposit…”. It is the deposit that may belong to a certain genetic type, not opposite.

Response: Thanks for your suggestions. We have completed the modification as follows: “The relative consensus amongst scientists is that the Yanjingou gold deposit belongs to an orogenic gold deposit, and ore types within it primarily consist of quartz vein and phyllic rock”.

  1. Lines 56-64. Given the previous studies, including a recent one, which considers an adjacent Hongqigou deposit of similar type, is there any conundrum and/or scientific significance in defining the orogenic type of gold occurrence in this study (as emphasized in the abstract)? I believe not. Besides, in the Introduction, the authors would benefit from providing some key features, which allowed to classify the mentioned deposits as orogenic.

Response: Yes, several previous researchers and we have also argued the opinion of an orogenic gold deposit for Wulonggou gold field, but the origins of ore-forming fluids and sources of metals responsible for gold mineralizations in the Shenshuitan gold deposit remain controversial. Orogenic gold deposits usually show similar geotectonic settings of orogeny but variable origins of ore-forming fluids and sources of metals (Groves et al., 1998; Goldfarbs et al., 2015). Therefore, in this paper, we report a systematic study of the H, O, S, and Pb isotope compositions of gold-bearing ores in order to characterize the origins of ore-forming fluids, sources of metals, and the ore genesis of the Shenshuitan gold deposit. We have revised the manuscript as mentioned above.

  1. Lines 302-315. I do not think there is a need to list all ranges of sulfur isotope composition for all sulfides in the text, given that there are several figures and a supplementary table with the original data. Instead, I would focus on highlighting the main similarities and differences, which are linked (or not) to types of mineral assemblages, their temperatures of precipitation etc.

Response: We have deleted all ranges of sulfur isotope composition for all sulfides in the text. And the main similarities and differences have been discussed in the revised manuscript as follows: “Pyrite has a highly variable range of δ34SVCDT values, while other sulfides yield a relatively narrow range, perhaps because pyrite is the most common sulfide, even in barren host rocks.” “Ore subtype 1 had highly variable δ34SVCDT values, likely because ore subtype1 was the most complicated mixture of ore-forming fluid and host rocks.” Detailed discussion should be found in the Section 5.2.

  1. Line 347 and below. Same as for S – you don’t just list the Pb/Pb ranges, instead show as straightforward as possible there is (or is not) a dependence between Pb isotopes and mineralogy of rocks/ores, including possible links to host rocks or any other isotopic sources of Pb, which are used in figures for interpretation.

Response: We also deleted the list of the Pb/Pb ranges of every sulfides in the text. And a dependence between Pb isotopes and mineralogy of ores has been discussed in the revised manuscript as follows “As with sulfur isotope, ore subtype 1 had highly variable Pb isotope ratios compared to others mostly because ore subtype1 was the most complicated mixture of ore-forming fluid and host rocks.” Detailed discussion should be found in the Section 5.3.

  1. Line 465 (and again – multiple times elsewhere in the text). I really doubt of calculating the average values in the ranges like these (e.g., from -9 to +8). This would not be a representative value in any case. For S, here I do not see the compromising explanation of S isotope variations based on mineralogical characteristics and tectonic history of the deposit. Why would the original fluid have the heaviest S?

Response: Our δ34S values ranging from −7.3‰ to 9.6‰ are broadly similar to some of the sediment-hosted orogenic gold deposits (e.g., −6.3‰ to 9.3‰, Chang et al., 2008). And Goldfarb et al. (2015) also argued that orogenic gold deposits are commonly characterized by extremely variable δ34S values, and reported δ34S values have been shown to be as low as −20‰ and as high as +25‰ for sulfide minerals from orogenic gold deposits. We have revised the Section 5.2 as mentioned above.

The reasons for the original fluid have the heaviest S are discussed as follows:  The large range (16.9‰) of δ34Ssulfide or δ34SH2S indicates a mixing source of sulfur. In particular, on the basis of the large difference between the ore subtype 1 in the granite and ore subtype 2 in the slate (Figure 9b, Figure 9d), we infer that the sulfur may have been mixed from a unique original ore-forming fluid and different host rocks (such as brecciated granite or slate). Low-grade metamorphism can release significant Au, S, and other elements from the source rocks to be subsequently concentrated by hydrothermal processes, forming orogenic gold deposits at higher crustal levels [84]. The sulfur isotope composition of the resulting S-bearing fluid in the Shenshuitan gold deposit may be similar to that of sulfides from quartz veins in the Hongqigou gold deposit which exhibited the heaviest δ34S values (Figure 9e, [29]). In summary, the original sulfur source of the ore-forming fluids of the Shenshuitan gold deposit may have the heaviest δ34S values similar to the Hongqigou gold deposit, and the different ore subtypes have variable S isotope compositions that may account for different degrees of water–-rock reaction/mixing and different host rock types.

  1. Line 522 and below. Again, for Pb isotopes, a few ephemeral isotopic reservoirs are listed, which might contribute to isotopic variations (which are pretty large), but no particular mechanisms were provided to account for mixing between them.

Response: We have discussed mechanisms of mixing for Pb isotopes in detail as follows: The field for the Pb isotope composition of sulfides in the Shenshuitan gold deposit overlapped partly with that of the Hongqigou gold deposit (purple dashed line in Figure 11a and Figure 11b) or potassic feldspars in the granite host rock (yellow dashed line in Figure 11a and Figure 11b). The sulfides in quartz veins from the Hongqigou gold deposit yielded relatively invariable Pb isotope compositions, which is consistent with no apparent mixing between ore-forming fluids and host rocks. Given that the ore-forming fluids in the Shenshuitan gold deposit are the same as those in the Hongqigou gold deposit as mentioned above, the Pb isotope compositions of sulfides in the Shenshuitan gold deposit would be similar to each other if no replacement happened during mineralization. However, phyllic rock ores which are products of strong water–rock reaction between ore-forming fluids and host rocks are actually dominant in the Shenshuitan gold deposit. Furthermore, the Pb isotope compositions of sulfides in the Shenshuitan gold deposit were roughly located between those of the Hongqigou deposit and those of the potassic feldspars in the granite wall rock (Figure 11a, Figure 11b), perhaps indicating that the former was a mixture between ore-forming fluids in the Shenshuitan gold deposit and granites. Moreover, the above three Pb isotope compositions mostly plotted within the age (215 Ma)-corrected sedimentary wall rocks in the uranogenic plot (gray dashed line in Figure 11a) and totally plotted within it in the thorogenic plot (gray dashed line in Figure 11b), if a few abnormal Pb isotope compositions falling outside the plot can be ignored. Therefore, host rocks of silicic slate also contribute to lead sources for sulfides in the Shenshuitan gold deposit. Strong water–rock reaction resulted in phyllic alteration and mineralization in ductile fault zones and near host rocks, as well as mixing of Pb isotopes.

  1. Line 596 and further. You state that the fluid was oxidized (CO2-bearing), but above, the reduced S is suggested as a major form in the fluid (Line 460). It is better to constrain the oxygen fugacity ranges, which are relevant for an initial fluid and ore precipitation process. What do you think of a major barrier for precipitation? (cooling/decompression and boiling/fluid reduction etc.)?

Response: Thank for your comments. Here we argued the ore-forming fluids in the Shenshuitan gold deposit have reduced state of the sulfur according to the sulfide assemblage. The fluid are CO2-bearing one which may not mean an oxidized state of the sulfur. As we known, there are generally significant amounts of H2S, assumed to be important carriers of the gold, in relatively consistent H2O–NaCl–CO2 ± CH4 ore-related fluid in most orogenic gold deposits(e.g., Goldfarb et al., 2015). Therefore, the reduced state of the sulfur is consistent with an ore-fluid redox state that is normally more reducing than the hematite-magnetite buffer (Phillips and Evans, 2004; Phillips and Powell, 1993). 

Sorry we did not do detailed studies of fluid inclusions for the Shenshuitan gold deposit, so we can’t answer in detail what is a major barrier for precipitation and the oxygen fugacity ranges. We guessed that decompression might be the major barrier for precipitation, because the association of orogenic gold deposits with major ductile faults in low- to medium-grade metamorphic environments has been explained by the fault-valve model [Siboson et al., 1988].

Kind Regards,

Qing-feng Ding

Reviewer 3 Report

The article is an important study of the Shenshuitan gold deposit located in the Eastern Kunlun Orogen (EKO) in Northwestern China, using the geochemistry of the H, O, S, Pb isotopes.

The authors make a special contribution to the source of the metals and fluids that generated the gold mineralization of Shenshuitan.

The measured values δ18O of quartz and sericite and the δD values of the fluid inclusions of quartz suggest that the metamorphic fluids that would have formed the ore from the Shenshuitan deposit could have been mixed with meteoric water.

Bulk values δ34S suggest that sulfur sources come from hydrothermal fluids but also from host rocks (granites and slates)

The situation is similar for lead isotopes which indicate as the source the hydrothermal fluids and the host rocks of the mineralization of Shenshuitan.

As a final conclusion, the authors suggest that the Shenshuitan gold deposit consists of hydrothermal fluids mixed with metamorphic fluids mixed with meteoric water associated with the final collision between the Bayan Har – Songpanganzi Terrane and the Eastern Kunlun Orogen.

Besides these achievements there are also some observations on the article:

  1. The photomicrographs in Figure 5 are not very clear because the polished sections are of low quality. Authors should obtain quality polished sections and make other photomicrographs to introduce in Figure 5.
  2. The authors classify Shenshuitan ore deposit as gold deposits. On page 5 lines 154-155 the authors say that there are bodies with an average gold grade of 6.74 ppm. The authors do not have any photomicrograph with native gold granules in pyrite and arsenopyrite (See those stated in line 170-171). If you have not discovered native gold granules then you should replace the title gold deposit with quartz vein or sulfides vein. Native gold is an indicator of gold deposit and you do not have such a photomicrograph. In order to keep the name of gold deposit, the native gold must be identified and a photomicrograph must be presented.

Author Response

Dear Reviewer 3,

Many thanks for your comments on our manuscript entitled "Isotope Geochemistry of the Shenshuitan Gold Deposit within the Wulonggou Gold Field in the Eastern Kunlun Orogen, Northwest China: Implications for Metallogeny". We have revised the manuscript according to your comments. Below are our detailed response to your comments:

  1. The photomicrographs in Figure 5 are not very clear because the polished sections are of low quality. Authors should obtain quality polished sections and make other photomicrographs to introduce in Figure 5.

Response: We are terribly sorry that we can not provide high quality polished sections and take other photomicrographs. Because we cut each polished section into several cylindrical shaped samples with a 3.3 mm diameter which were then mounted into a resin mount for in situ sulfur isotope analyses. But we have provided the new Figure 5 with larger purple fonts for symbols of the minerals and some high resolution photomicrographs.

  1. The authors classify Shenshuitan ore deposit as gold deposits. On page 5 lines 154-155 the authors say that there are bodies with an average gold grade of 6.74 ppm. The authors do not have any photomicrograph with native gold granules in pyrite and arsenopyrite (See those stated in line 170-171). If you have not discovered native gold granules then you should replace the title gold deposit with quartz vein or sulfides vein. Native gold is an indicator of gold deposit and you do not have such a photomicrograph. In order to keep the name of gold deposit, the native gold must be identified and a photomicrograph must be presented.

Response: Actually, there are few native gold granules in the Shenshuitan gold deposit which were reported by First Institute of Qinghai Geology Survey(FIQGS, 2010). Unfortunately, we did not found any native gold granule in our studied polished sections. We have inserted a photomicrograph with a native gold granule after FIGQS (2010) into the new Figure 5.

Kind Regards,

Qing-feng Ding

Round 2

Reviewer 1 Report

Dear authors,

Thanks for making all revisions on a draft. 

All the best

Author Response

Dear Reviewer 1,

Thanks.

Kind Regards,

Qing-feng Ding

Reviewer 3 Report

The authors did not make the requested changes.
The photomicrographs in Figure 5 are still blurry because the polished sections are not quality. Those in the previous version have not been replaced.
Photomicrograph 5i in Figure 5 is very blurry and is a copy of an exploration report, so it is not the authors' contribution.

Please revise figure 5 according to my requests.

Author Response

Dear Reviewer 3,

Many thanks for your comments on our manuscript entitled "Isotope Geochemistry of the Shenshuitan Gold Deposit within the Wulonggou Gold Field in the Eastern Kunlun Orogen, Northwest China: Implications for Metallogeny". We have revised the manuscript according to your comments. Below are our detailed response to your comments:

  1. The authors did not make the requested changes.The photomicrographs in Figure 5 are still blurry because the polished sections are not quality. Those in the previous version have not been replaced. Photomicrograph 5i in Figure 5 is very blurry and is a copy of an exploration report, so it is not the authors' contribution. Please revise figure 5 according to my requests..

Response: We have tried our best to revise the Figure 5 in this round of revision. All photomicrographs in Figure 5 are newly captured from polished sections of resin mounts which consist of 6 cylindrical shaped samples with 3.3 mm-diameter for in situ sulfur isotope analyses, as we have no any more polished sections of ores. Few shadows that can be seen in some photomicrographs in Figure 5 are resin. We hope this modification can meet your requirement.

Kind Regards,

Qing-feng Ding

Round 3

Reviewer 3 Report

I agree with the publication of the article

Author Response

Thanks.